# Enhancing Efficiency of Natural Product Structure Revision: Leveraging CASE and DFT over Total Synthesis

**DOI:** 10.3390/molecules28093796

**Published:** 2023-04-28

**Authors:** Mikhail Elyashberg, Sriram Tyagarajan, Mihir Mandal, Alexei V. Buevich

**Affiliations:** 1Advanced Chemistry Development Inc. (ACD/Labs), Toronto, ON M5C 1B5, Canada; 2Medicinal Chemistry, Merck & Co., Inc., Kenilworth, NJ 07033, USA; sriram_tyagarajan@merck.com (S.T.); mihirbaran.mandal@merck.com (M.M.); 3Analytical Research and Development, Merck & Co., Inc., Kenilworth, NJ 07033, USA

**Keywords:** natural products, structure revision, NMR, CASE, DFT

## Abstract

Natural products remain one of the major sources of coveted, biologically active compounds. Each isolated compound undergoes biological testing, and its structure is usually established using a set of spectroscopic techniques (NMR, MS, UV-IR, ECD, VCD, etc.). However, the number of erroneously determined structures remains noticeable. Structure revisions are very costly, as they usually require extensive use of spectroscopic data, computational chemistry, and total synthesis. The cost is particularly high when a biologically active compound is resynthesized and the product is inactive because its structure is wrong and remains unknown. In this paper, we propose using Computer-Assisted Structure Elucidation (CASE) and Density Functional Theory (DFT) methods as tools for preventive verification of the originally proposed structure, and elucidation of the correct structure if the original structure is deemed to be incorrect. We examined twelve real cases in which structure revisions of natural products were performed using total synthesis, and we showed that in each of these cases, time-consuming total synthesis could have been avoided if CASE and DFT had been applied. In all described cases, the correct structures were established within minutes of using the originally published NMR and MS data, which were sometimes incomplete or had typos.

## 1. Introduction

Each year, a significant number of natural products are isolated worldwide [1]. The primary purpose of isolating natural compounds is to increase the number of biologically active molecules that can be used as the foundation for creating new drugs. Once these compounds are isolated and purified, they undergo in-depth characterization of their biological activity and molecular structure. Spectroscopic techniques, such as NMR, MS, UV-IR, ECD, VCD, etc. are commonly used to determine their structure. Despite significant progress in structure elucidation methods in recent decades, structural misassignments still occur frequently (see reviews [2,3,4,5,6,7]). The cost of such misassignments is significant. Firstly, the resources and time invested in the synthesis of the wrong molecule are usually wasted, and secondly, researchers must restart the synthesis process while also determining the correct structure of the desired molecule.

Revising a structure is a difficult task that usually requires an authentic sample. Researchers must repeat the entire structure elucidation process based on collecting a new set of spectroscopic data. If an authentic sample is unavailable, researchers must synthesize all plausible candidates with the hope that one of those will have identical spectroscopic characteristics to the authentic sample. The latter is known as the proof by the total synthesis, and is considered the strongest evidence of the authentic structure. While these structural misassignments provided synthetic chemists the opportunity to showcase their creativity and build skill in multistep synthesis, they often had been cumbersome and taken away valuable time from accomplishing the main objective—to probe and expand the biological properties of the natural products and their close analogs.

The solution to the structure revision problem can be found in the application of artificial intelligent (AI) technologies, which have been developed for the last three decades by a number of groups [8,9,10,11,12,13,14]. This branch of science is often called Computer-Assisted Structure Elucidation (CASE). Modern CASE algorithms are based on two major processes: (a) high efficiency of structure generation; and (b) high accuracy and efficiency of prediction of NMR chemical shifts for generated structures. The efficiency of structure generation, among other factors, is attained by the constraints derived from experimental data. In turn, the accuracy and efficiency of chemical shift predictions is achieved by empirical methods, including AI methodologies, trained on a large set of experimental values. Therefore, for successful CASE analysis, one would need to provide a set of NMR chemical shifts and a table of correlations observed in COSY and HMBC spectra. The CASE algorithms not only provide a logical deduction of all possible structures without exception that satisfy the experimental spectra and knowledge of the system, but also offer aids to assess the probability of structures. Ultimately, the program produces the most plausible structure, and if several equally probable structures are detected, the final choice can be made based on the prediction of NMR chemical shifts using more accurate methods of quantum chemistry, such as density functional theory (DFT) calculations [15,16]. It has been shown that adding DFT methods to CASE analysis often enhances the overall robustness of structure selection and expands the applicability of CASE for solving stereochemical problems [15].

The power of CASE solutions for structure revision lies in their ability to use the original spectroscopic data published for misassigned structures. This eliminates the need for new spectroscopic data acquisition and enables the confirmation of the structure without the need for time-consuming total synthesis.

To illustrate the capabilities of CASE analysis for the structure revision of natural products, we examined twelve real cases where structure revisions of natural products were performed using total synthesis. To achieve this, the expert system ACD/Structure Elucidator [17] (ACD/SE) was utilized. The results showed that in each of these cases, total synthesis could have been avoided if CASE and DFT had been applied. The originally proposed incorrect structures were promptly detected, and then the correct structures were established within minutes using the originally published NMR and MS data when empirical chemical shift predictions were used, or within a couple of hours if chemical shifts were computed using DFT methods. It is noteworthy that CASE analysis was successful even when the original data sets were incomplete or contained misassigned chemical shifts.

## 2. Results

### 2.1. Macahydantoin B

Qiu and co-workers [18] isolated from the roots of *Lepidium meyenii* two similar compounds, macahydantoins A and B, having a novel 1,3-diazabicyclo[3.3.1]-nonane core appended with a benzyl moiety. The structures were originally elucidated using 1D and 2D NMR spectra. The structure of macahydantoin A was confirmed by synthesis, which allowed the authors to conclude that the structure of macahydantoin B (**1**), similar to macahydantoin A, is also correct.

Zhou and co-workers [19] isolated a natural product, macahydantoin C, whose skeleton differs from **1** only by lacking the methoxy group. However, it turned out that the NMR spectra of its non-benzyl moiety differed significantly from the spectra of **1**. To clarify the reason for the difference, structure **1** was synthesized. It was revealed that the NMR spectra of the synthetic structure differed from those published by Qiu and co-workers [18]. In addition, a four-bond HMBC correlation between H-6 proton to C-3 carbon was observed while the expected correlation between H-7 proton and C-1 carbon was absent. These observations led the authors [19] to hypothesize that structure **2** was correct. The proposed revised structure **2** was confirmed by total synthesis.



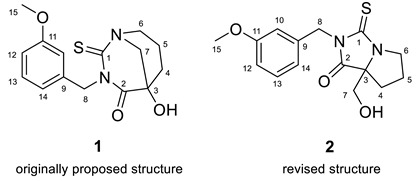



To verify the originally proposed structure **1**, we entered its structure and ^13^C NMR chemical shifts, as assigned by authors [18], into the ACD/SE CASE program. Chemical shift prediction was performed using three common methods: HOSE-code based (A), Neural Networks (N), and Incremental (I) (see Section 4 for details). In Figure 1, almost all carbons on the right side of the molecule have significant deviations between predicted and experimental values (marked in yellow), indicating the need to check the structure’s correctness. Consistent with these warning signs, both the average and maximum deviation of 7.76 ppm are somewhat larger than usually expected.

^13^C NMR chemical shifts [18], and HMBC and COSY correlations (Appendix A) available from [18] (a corrigendum published by the Qiu group) were entered to the CASE program, and a MCD (Molecular Connectivity Diagram) was created (Figure 2).

CASE analysis was initiated from the MSD (Figure 2) using the Fuzzy Structure Generation (FSG) algorithm [20] limiting the number of non-standard correlations to 1. Results of CASE analysis: k = 1987 → (structure filtering) → 161 → (duplicate removal) → 124, *t_g_* = 6 s, where k is the number of structures, *t_g_*—a time of structure generation. Then, 124 final structures were rank ordered based on ^13^C chemical shift predictions. Four top-ranked structures are shown in Figure 3.

As seen from Figure 3, the first-ranked structure coincides with the revised structure **2** proposed and synthesized by Zhou and co-workers [19]. This structure has the smallest deviations and its DP4_A_(^13^C) value of 99.99% confirms with high confidence that the CASE-proposed structure is correct. It is worth noting that the structure elucidation was performed in automatic mode (no user intervention) and it took just several seconds of computational time to revise the structure of macahydantoin B based on the original NMR and MS data. 

### 2.2. Clionastatin

Thousands of naturally occurring steroidal derivatives are known, but halogenated ones are extremely rare. Fattorusso et al. [21] were the first to analyze the burrowing sponge *Cliona nigricans* and discovered two new steroidal derivatives, clionastatins A (**3**) and B (**4**), which were isolated as the main components responsible for the cytotoxic activity. These molecules represent the first polyhalogenated steroids found in a natural organism, either marine or terrestrial, and can be regarded as the first examples of halogenated androstanes in nature.



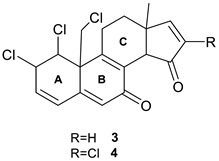



The structures of both compounds were determined using 1D and 2D NMR, and MS spectra. Fattorusso et al. [21] determined the relative stereochemistry of **3** and **4** by ROESY correlations, but were unable to obtain an X-ray crystal structure from the small amount of amorphous solid isolated. To confirm the novel structures of **3** and **4**, Tartakoff and Vanderwal [22] carried out synthesis of the ABC tricycle included in the structures. They synthesized the highly chlorinated ABC tricyclic ring system of clionastatins A and B in 12 steps, and showed that the key resonances from the A and B rings match nearly perfectly. The general conclusion was that the chemical shifts of the ABC tricycle differ only slightly from the spectrum of the natural products.

To ensure that structure **3** is unique and that no other structures correspond to the NMR spectra represented by Fantarusso and colleagues (Appendix A), we entered spectroscopic data from Appendix A into the ACD/SE program. The Molecular Connectivity Diagram is shown in Appendix A. Checking the MCD for the absence of contradictions in the 2D NMR data revealed that they contained at least one non-standard correlation. Therefore, we launched a fuzzy structure generation with automatically determined options. The result of the search was k = 2 → 2 → 1 and *t_g_* = 1 s. Thus, the only structure that was retained in the output file was structure **3** (Figure 4).

As observed in Figure 4, during the process of fuzzy structure generation (FSG), a non-standard HMBC correlation between H1 and C4 was detected. Additionally, the average deviations of d_A_ and d_N_ were slightly higher than the usual values that are characteristic of correct structures. These deviations may be due to the uniqueness of the structure of clionastatin A, as there are no close analogs of that structure in the ACD/SE database. However, the CASE analysis revealed that structure **3** is the only possible structure that can be deduced from the original spectroscopic data. 

### 2.3. Pyrostatin B (Ectoin)

Aoyama et al. [23] investigated *Burkholderia plantarii*, a bacterial pathogen of rice, and isolated two antibacterial compounds, pyrostatins A (**5**) and B (**6**), which bear a new 2-iminopyrrolidine carboxylic acid structure. Follow-up studies have shown that these compounds can be considered as potential therapeutic agents. In their continued search for bioactive molecules, Jiménez and co-workers [24] isolated and characterized a natural product from *Cliona tenuis* whose structure coincided with pyrostatin B. However, the spectral data for that compound were different from the reported spectral data of the compound reported as pyrostatin B [23]. Jiménez and co-workers [24] performed total synthesis of the isolate to confirm its proposed structure and demonstrated that the reported structure of pyrostatin B was incorrect. Furthermore, the search of the literature by Jiménez and co-workers [24] revealed that the actual NMR data reported for pyrostatin B matched those of ectoine (**7**), while pyrostatin A should be revised to 5-hydroxyectoine (**8**).



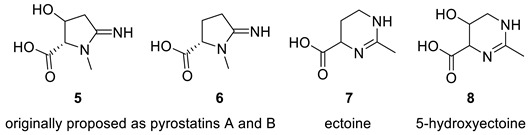



The verification of structure **6** with published NMR data for pyrostatin B using ACD/SE revealed large averaged and maximum ^13^C chemical shift deviation inconsistent with the proposed structure **6** (Figure 5).

To determine the correct structure consistent with the published data for compound **6**, the 1D and 2D NMR data [23] (Appendix A) were uploaded to ACD/SE, and a molecular connectivity diagram was automatically generated by the program (see Appendix A). However, the program determined that structure **6** could not be generated from this molecular connectivity diagram because the methyl group at 18.90 ppm was assigned the label “fb” (forbidden). This is because a methyl group with a chemical shift of 18.90 ppm cannot be connected to a heteroatom. The structure generation ended with the following results: k = 18 → 16 → 16, *t_g_* = 1 s. The two top-ranked structures are shown in Figure 6.

As seen in Figure 6, the best structure generated by the program is identical to the revised structure **7** determined by Jiménez and co-workers [24]. The second-best structure in the output file is a tautomer of structure **7**. As mentioned earlier, the originally proposed structure was not even generated due to a violation of one of the basic criteria of the program.

### 2.4. Madurastatin C

The siderophores madurastatins A1, B1, and MBJ-0035 are secondary metabolites used by pathogenic bacteria for taking up essential minerals. Madurastatin C1 was isolated by two different groups. First, in 2012 by Mazzei et al. [25] from the fermentation broth of *Actinomadura* sp. DSMZ 13491, and then in 2014 by Kawahara et al. [26] from the culture of *Streptosporangi* sp. 32552. The structure of the isolated madurastatin C [25] and MBJ-0034 [26] were identical and, in both cases, were assigned to structure **9** based on MS, 1D, and 2D NMR spectroscopic data. Interestingly, madurastatins C1 isolated by Hall et al. [27] was spectroscopically different from the originally proposed structure **9**. Recently, Thorson and Shaaban [28] proposed that an N-terminal 2-(2-hydroxyphenyl)-oxazoline ring instead of an N-terminal aziridine ring is present in madurastatin siderophores. To test this hypothesis, a salicylate-containing fragment of **9** was synthesized as both an aziridine (**10**) and as an oxazoline (**11**) for comparison. Comparison of the ^1^H and ^13^C NMR of the authentic sample of **9** with synthetic compounds **10** and **11** confirmed Thorson and Shaaban’s revised structure of madurastatin C1 (**12**), which has an oxazoline moiety.



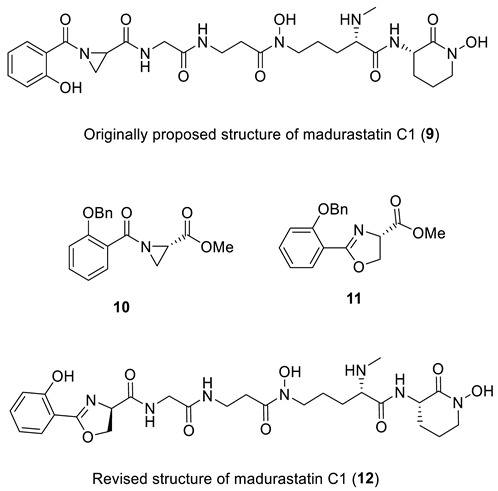



Verification of the originally proposed structure of madurastatin C was done by the ACD/SE program using published 1D and 2D NMR spectroscopic data [26] (see Appendix A). Elevated average deviations and unacceptably large maximum deviation for structure **9** clearly showed that the structure was incorrect (Figure 7).

Automated structure analysis was accomplished using the same set of chemical shifts, and MCD was created by the program based on COSY and HMBC correlations (Appendix A). To ensure that the original structure would also be generated, structural filtering was switched off during the analysis. The structure generation ended with the following results: k = 6672 → 712, *t_g_* = 2 m. The three top-ranked structures are shown in Figure 8.

As is clear from the data in Figure 8, the revised structure **12** is the top-ranked, and is statistically well-separated from other candidates. The original structure was placed on the 28th position. It is noteworthy that not only was the N-terminal 2-(2-hydroxyphenyl)-oxazoline ring correctly predicted by the ACD/SE program, but so were all remaining parts of the molecule.

### 2.5. Dichomitol

Dichomitol, a novel sesquiterpenoid natural product, was first isolated from *Dichomitus squalens*, a commonly found white-rot *Basidiomycete fungus*, in 2004 by Huang et al. [29]. Based on 1D and 2D NMR data, they proposed structure **13**. In 2006, Mehta and Pallavi [30] achieved total synthesis of this proposed structure and observed significant differences between the synthesized and proposed structures. They suggested that the spectral data of the natural product should be reinvestigated, but no structure revision was done. In 2011, Wei et al. [31] isolated a series of new sesquiterpenes from the mycelial solid cultures of *Dichomitus squalens*, among which was a compound with identical NMR spectra to those published for dichomitol. They reanalyzed the NMR spectra and proposed protoillud-6-ene-8b,13,15-triol (**15**) as the revised structure of dichomitol.



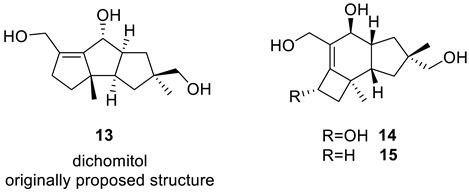



The ^13^C chemical shift prediction for structure **13** revealed that the average deviations did not indicate any errors in the structure (Figure 9). However, five-membered ring carbons exhibited chemical shift predictions outside the acceptable range for four out of five carbons (highlighted in yellow in Figure 9). This finding justifies further analysis using the CASE method.

Entering the 1D NMR chemical shifts and key HMBC correlations (Appendix A) from the original paper on dichomitol [29] into ACD/SE, creating an MCD (Appendix A), and running strict structure generation did not detect any contradictions in the HMBC data, and generated only one structure identical to **13** in no time. To verify the solution’s stability and identify other possible structures, fuzzy structure generation was initiated, admitting the presence of one NSC. This resulted in saving 28 structures in 0.2 s, with structure **13** ranking as the best again. Next, FSG was carried out with the assumption that two possible NSCs of unknown length may exist in the generated structures. The solution for that generation was as follows: k = 2208 → 453 → 269, *t_g_* = 2 s. Figure 10 shows the three top-ranked structures.

After performing the procedure to check solution stability, we obtained the revised structure **15** as the most probable one, characterized by the lowest average and maximum chemical shift deviations and DP4 parameters close to 100%. Furthermore, we corrected the ^13^C chemical shift assignment performed in [31]: atoms C6 (129.1) and C7 (145.8) swapped places in structure **13**. Finally, the two non-standard correlations assumed in the CASE analysis are consistent with the relatively large *J*(C,H) couplings predicted by DFT computations: ^4^*J*(C3,H8) = −1.35 Hz, ^4^*J*(C5,H15) = −1.46 Hz, and ^4^*J*(C5,H15′) = −1.68 Hz [32].

To demonstrate the synergistic power of the CASE and DFT methods, we also conducted a stereochemical analysis of dichomitol. The molecule of dichomitol (**15**) contains five chiral centers, resulting in 32 possible stereoisomers, including 16 pairs of enantiomers. As NMR spectroscopy can determine only relative stereochemistry, we generated one set of enantiomers with fixed R-chirality at the C11 carbon (Figure 11). For each stereoisomer in Figure 11, we performed a conformational analysis using MMFF94 force-field calculations with the Spartan′20 program. We then optimized the generated ensembles of conformations and calculated chemical shifts at the DFT level of theory using the Gaussian16 program. The averaged ^13^C and ^1^H chemical shifts were obtained using a Boltzmann distribution based on electronic DFT energies for each stereoisomer of **15**. The RMSD values between the DFT-predicted and experimental chemical shifts are shown in Figure 12. As seen in Figure 12, the natural isomer of dichomitol (**15a**) has the lowest ^1^H and ^13^C RMSD values, indicating that DFT-computed chemical shifts can differentiate the correct stereoisomer from all possible isomers (for more details see Appendix A).

### 2.6. Samoquasine A

In 2000, Morita et al. [33] isolated the cytotoxin natural product samoquasine A from the seeds of *Annona squamosa* and proposed its structure as structure **16** based on spectroscopic and chemical evidence. However, in 2002, the same group retracted their structural assignment and reported that samoquasine A was identical to perlolidine (**17**). They did not specify what led them to this conclusion. In 2003, Yang and co-workers [34] synthesized compound **16** and found that its spectra did not match those published in [33]. They proposed structure **18** as the true structure of samoquasine A. To verify this hypothesis, Monsieurs et al. [35] synthesized compound **18** in 2007, but the NMR spectra differed from those published by Morita et al. [33] Still chasing the true structure of samoquasine A, Timmons and Wipf [36] performed DFT predictions of ^13^C NMR chemical shifts of 48 isomeric compounds in 2008 and discovered that samoquasine A was indeed identical to perlolidine (**17**). Finally, in 2018, Piggot and co-workers [37] confirmed this conclusion by directly synthesizing compounds **17** and **18** using a novel route involving a β-selective Heck reaction of butyl vinyl ether.



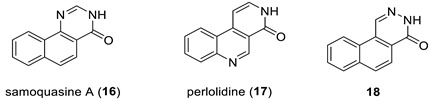



We will show that the 18-year saga about the true structure of samoquasine A could be solved in a matter of seconds if only the authors of [33] had used CASE in their work. First, we verified the ^13^C chemical shifts of structure **16** (Figure 13a), which showed a substantial deviation, indicating an incorrect structure. Next, we entered the 1D NMR and HMBC data presented in [33] (Appendix A) into the ACD/SE program. A logical analysis of the data presented on the MCD (Appendix A) showed that there were no contradictions. However, strict structure generation produced zero results, revealing the presence of non-standard correlations in the HMBC spectra. Therefore, we used fuzzy structure generation, which resulted in k = 182 → 94 → 73, *t_g_* = 4 s. The top three structures of the output file are shown in Figure 14.

Figure 14 clearly shows that the program provided the unambiguous establishment of the correct structure **17**, whereas structure **18** was found in the ranked output file at the 38th position (Figure 13b). CASE analysis convincingly demonstrated that this structure must be rejected and that the revised structure of samoquasine A was identical to that of perlolidine (**17**). Notably, the only non-standard correlation that was assumed in the fuzzy generation mechanism was well-grounded. DFT computations confirmed that the corresponding ^4^*J*(C,H) coupling between the C6 carbon and H4 proton is 0.78 Hz, consistent with an observable peak in HMBC spectra [32]. Thus, the structure elucidation, which began in 2000 and spanned nearly two decades, could have been almost instantly established using CASE.

### 2.7. Palmarumicin B6

Shan et al. [38] isolated nine new spirobisnaphthalenes, palmarumycins B1–B9, from the endophytic fungus *Berkleasmium* sp. The structures of the new compounds were elucidated by analyzing their 1D and 2D NMR and high-resolution ESI-MS spectra and by comparison with known compounds. Palmarumycin B6 exhibited larvicidal and antibacterial activity, and its structure (**19**) was originally proposed based on NMR and MS data analysis. However, when Liu and co-workers [39] synthesized the proposed structure (**19**), they found that its ^1^H and ^13^C NMR data were not consistent with those earlier reported for palmarumycin B6. They hypothesized that the actual structure of palmarumycin B6 was isomeric structure (**20**). To verify this hypothesis, they synthesized structure **20** and confirmed its correctness by X-ray diffraction analysis. The ^1^H and ^13^C NMR data of **20** were identical to those reported for palmarumycin B6, which led to the revision of its structure.



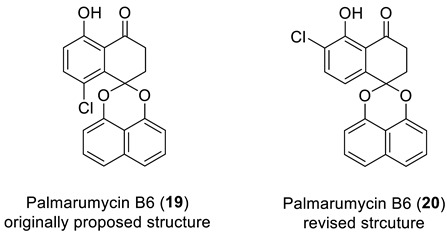



Structure verification of the originally proposed structure of palmarumycin B6 (**19**) was done based on the published ^13^C NMR chemical shifts using empirical chemical shift prediction within the ACD/SE program (Figure 15). Although the average deviations of predicted carbon chemical shifts are within acceptable values, the maximum deviation of 18.71 ppm for a carbon atom with 139.6 ppm (marked in red in Figure 15) indicates that structure **19** is questionable and requires further verification.

Unfortunately, the authors of the works [38,39] did not publish tables containing 2D NMR data. However, the program provides a special procedure, which can be conditionally called “cutting out the green fragment,” for cases where 2D NMR spectra are not available. The program generates a “reduced” MCD for the fragment that has deviations less than 3 ppm (marked in green in the structure verification figure, such as Figure 15) and uses that MCD in full structure analysis. This approach turns out to be very effective in cases where fragments with correct structures are found in the tested molecular structures, limiting the search only for less defined fragments of the structure.

From the structure with the assigned ^13^C and ^1^H chemical shifts presented in Figure 15, fragment **19a** (a “green fragment”) containing carbons, for which chemical shifts were predicted with good accuracy, was “cut out”.



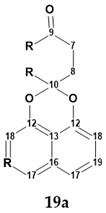



Next, an MCD was created (Appendix A) containing this fragment and free atoms within the molecular formula C_20_H_13_O_4_Cl. The structure generation gave the following result: k = 13,524 → 1050 → 209, *t_g_* = 30 s. Manual inspection of structures in the output file showed that many structures contained unlikely cyclobutene fragments. Those structures were filtered out, after which 61 structures remained in the output file. The six top-ranked structures are shown in Figure 16.

As can be seen from Figure 16, the revised structure **20** is characterized by the lowest average and maximum deviations and by high empirical probabilities DP4. However, the CASE analysis improved the assignments of chemical shifts in original structure and reassigned C2 and C4 carbons, which led to significant reductions of its chemical shift deviation (compare data in Figure 15 and Figure 16). As a result of that correction, the original structure was also quite plausible. 

To further strengthen our conclusion that the revised structure **20** (#1 by CASE ranking) was the correct one, we conducted a DFT analysis of the top six CASE candidates, which included the originally proposed structure **19**. As seen from Table 1, all statistical parameters of DFT-predicted chemical shifts (RMSD, max_dev, and *r*) are clearly in favor of the revised structure **20** (see Appendix A for more details).

### 2.8. Nocarbenzoxazole G

Sun and coworkers [40] isolated seven new benzoxazoles, nocarbenzoxazoles (A–G) derivatives from the halophilic strain *Nocardiopis lucentensis* DSM 44,048 and elucidated their structures using 1D and 2D NMR spectroscopic data and HR-ESI-MS. They were assayed for their cytotoxicity against a panel of human tumor cell lines. Nocarbenzoxazole G was the only one which showed selective activity against some of them (HepG2 & HeLa).

Kim et al. [41] synthesized nocarbenzoxazole F and G via a Pd-catalyzed directed arylation of 2-H benzoxazole with the corresponding aryl bromides using microwave radiation followed by reduction of the methyl esters. The structures of the products were determined by 1D and 2D NMR methods. While the data for synthetic norbenzoxazole F (**21**) matched the reported natural norbenzoxazole F, the data for the synthetic norbenzoxazole G was found to be different from that of the natural product (**22**).



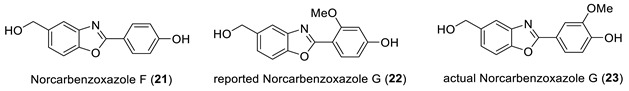



The structure of norcarbenzoxazole G was revised by total synthesis of structure **23**, followed by matching its NMR chemical shifts to those reported for natural norbenzoxazole G [40].

Structure verification of the originally proposed structure of norcarbenzoxazole G (**22**) was done using ACD/SE predictor and ^13^C and ^1^H NMR chemical shifts available from [40] (Figure 17 and Appendix A).

As can be seen from Figure 17, the original structure **22** is most likely incorrect.

Unfortunately, neither the original article [40] nor the followed-up work [41] contain 2D NMR spectra, making the application of the ACD/SE program in standard mode impossible. In this case, the green bicyclic fragment on the left side of the molecule was “cut out” by the user. Next, the program automatically created the MCD containing this fragment (Appendix A). The structure generation was performed with that MCD, resulting in the following output: k = 552 → 72 → 22, *t_g_* = 4 s. The three top-ranked structures from the output file are shown in Figure 18.

We see that the revised structure was placed in the second position, while the original structure was at the 14th (not shown here). DP4 probabilities were calculated for all three methods of chemical shift prediction, and two of them (DP4_N_ and DP4_I_) selected the revised structure as the most probable one. To further support our conclusion that the revised structure **23** (#2 by CASE ranking) is correct, we conducted a DFT analysis of the top three CASE candidates and the originally proposed structure **22**. The RMSD, max_dev, and *r* parameters of the DFT-computed chemical shifts provide clear evidence for the correctness of the revised structure **23** (see Table 2 and Appendix A).

### 2.9. Hetiamacin A

Sun and co-workers [42,43] isolated hetiamacins A-D from the cultured broth of *Bacillus subtilis*. The structure of hetiamacin A, a member of this amicoumacin group of antibiotics, was elucidated by various spectrochemical methods, but the stereochemistry of all five chiral centers was not determined. It was hypothesized by Wu et al. [44] that all five chiral centers in hetiamacin A preserved the (S)-configurations as AI-77-B, a potent gastroprotective agent, as they have similar structures.



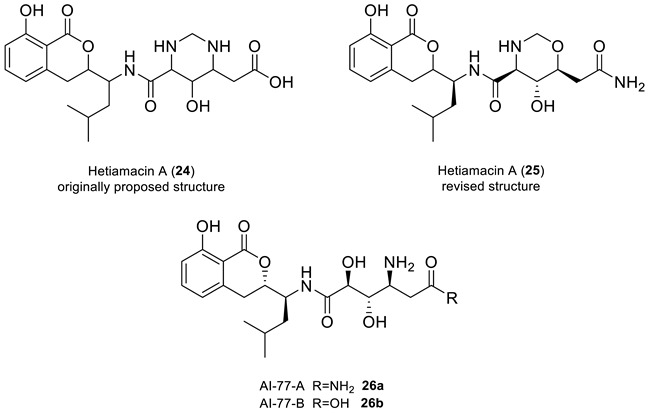



A multi-step total synthesis of hetiamacin A from AI-77-B (**26b**), achieved through the cyclization of the 8′-hydroxyl group and the 10′-amine group (using a key Witting olefination and Sharpless asymmetric dihydroxylation reaction), revealed that the originally reported structural assignment for hetiamacin A (**24**) was incorrect, as confirmed by the NMR data of the synthetic compound [44]. Based on their synthetic work, the authors were able to elucidate the stereochemistry of all five chiral centers in hetiamacin A, and revised its original reported structure to structure **25**.

First, structure verification of the originally proposed structure **24** [42] was carried out using ACD/SE. Both average and maximum deviations of predicted ^13^C chemical shift for structure **24** (see Figure 19) exceed values common for correct structures, clearly indicating that the structure was assigned erroneously.

In article [44], where structure **24** was revised, only 1D NMR chemical shifts were presented for the revised structure **25**, which made it impossible to use experimental 2D NMR spectra for structure generation. For such situations, ACD/SE provides an option called “create project for structure”, which allows the creation of an MCD containing all theoretically possible HMBC-based connectivities corresponding to structure **24** (see Appendix A). The structure generation from this MCD was completed with the following results: k = 172,800 → 60 → 60, *t_g_* = 3 m 15 s. We found that sixty structures were in agreement with theoretical HMBC correlations. Figure 20 shows the three top-ranked structures by CASE analysis. The structure ranking procedure placed the revised structure **25** to the first position, while the proposed structure **24** was ranked 17th in the output file. The average chemical shift deviations for the top three structures were very similar. To further differentiate the revised structure **25** (#1 by CASE ranking) from the closest candidates, we performed a DFT analysis of the top three CASE candidates and the originally proposed structure **24**. The RMSD, max_dev, and *r* parameters of the DFT-computed chemical shifts clearly demonstrate the correctness of the revised structure **25** (see Table 3 and Appendix A).

Empirical and DFT chemical shift calculations have distinguished a revised structure in this case. However, this approach cannot be recommended as a routine tool for CASE-based structure revision since it is only successful under specific conditions.

### 2.10. Uniflorine A

Matsumura and co-workers [45] isolated two novel compounds, uniflorines A (**27**) and B (**28**), and a known triol (**29**), water soluble alkaloids, from a Paraguayan natural medicine, Ñangapiry, isolated from the leaves of tree *Eugenia uniflora* L. These were used as an antidiabetic agent. Using 1D and 2D NMR data, the compounds were characterized, and compound **28** was found to be an isomer of **27**.



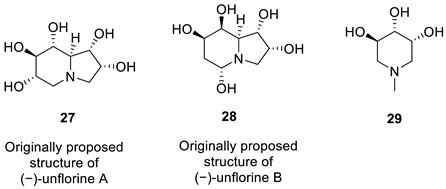



The inhibition by the three compounds was found to be dose-dependent and comparable to those of Acarbose.

Davis et al. [46] undertook a nine-step, resource- and time-consuming synthesis of the uniflorine A from L-xylose. They established the structure of synthetic **27** by X-ray crystallography. To their surprise, ^1^H and ^13^C NMR spectra did not match the reported literature spectra, and they concluded that the structure originally assigned to uniflorine A was incorrect. Ritthiwigrom et al. [47] noticed the similarity in NMR spectra of uniflorine A and a known alkaloid casuarine (**31**). They hypothesized that uniflorine A was 6-epi-casuarine (**30**), which they proved by a nine-step synthesis of its enantiomer, (+)-uniflorine A, from D-xylose.



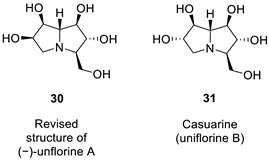



We started the analysis of the uniflorine A structure by chemical shift verification of published ^13^C chemical shifts with ACD/SE. The predictions of ^13^C chemical shifts for the originally proposed structure of uniflorine A (**27**) showed relatively large average deviation and max_dev parameters (Figure 21). Four out of eight carbons in the proposed structure of uniflorine A had significant deviations (marked in yellow in Figure 21). These calculations clearly indicated the potential problem with the proposed structure of uniflorine A. 

Because the 2D NMR spectra were not reported in the originally published paper on uniflorine A [45], the CASE analysis was initiated by constructing the MCD using the proposed structure **27** with the command “create project for structure.” By doing so, the ^13^C and ^1^H chemical shifts and presumed HSQC data were transferred from structure **27** to the MCD (Appendix A). Structure generation was completed with the following results: k = 6,674,400 → 10,296 → 13, *t_g_* = 6 m 30 s. When the structures containing oxetane ring, rare in natural products, were removed from the output file, only three structures were left in the resulting file (Figure 22).

As shown in Figure 22, the CASE analysis of the published data correctly predicted the overall scaffold of the revised structure **30** proposed by Ritthiwigrom et al. [47], despite the limited spectroscopic data available for uniflorine A. Given that the compound’s relative stereochemistry could be determined from NOE data at the time of isolation, it seems evident that the entire structure revision by direct synthesis could have been avoided if the CASE program had been utilized.

### 2.11. Altechromone A

Königs and co-workers [48] set out to synthesize related analogs of altechromone A [49], a chromone derivative which shows biological activity. They devised a route employing aldol reaction of intermediate silyl ether analog, **33**. However to their surprise, the NMR spectroscopic data of **34** showed to be different from the previously reported structure of altechchromone A, isolated naturally [49]. They also synthesized isomeric congeners **35**, in addition to an analogous coumarin derivative **36**, to rule out a coumarin architecture.



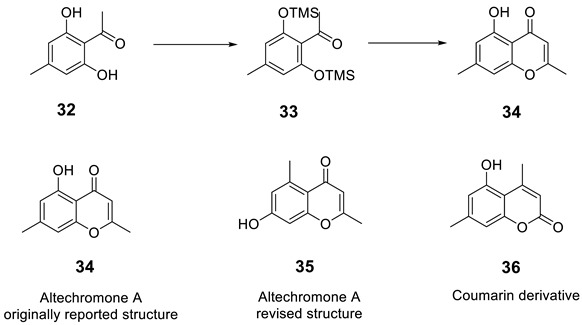



The best fit for the spectroscopic data with the isolated compound turned out to be compound **35**. The spectroscopic data and melting point for the synthesized **34** did not fit the structure previously shown. The structure of altechromone A was revised and reassigned as the one shown as **35**.

We began the analysis of altechromone A’s structure by verifying the published ^13^C and ^1^H chemical shifts with ACD/SE. We inputted the ^13^C and ^1^H NMR data (see Appendix A) and structure **34**, along with its NMR chemical shift assignments, into ACD/SE and carried out chemical shift predictions (see Figure 23). 

Figure 23 shows that the originally proposed structure **34** for altechromone A has larger chemical shift deviations, and therefore, the correctness of this structure should be verified. As COSY and HMBC data were not available from articles [48,49], the option to create MCD from the original structure using the “create project for structure” command was employed, similarly to the cases discussed above. The structure generation was performed from the MCD (Appendix A), resulting in the following: k = 396,696 → 866 → 42, *t_g_* = 2 m 20 s. The top three ranked structures of the output file are presented in Figure 24.

The CASE analysis of the originally published 1D NMR data was able to unequivocally distinguish the revised structure **35**. It is noteworthy that the synthesis of coumarin isomer **36** could also be avoided, as its chemical shifts, according to ACD/SE chemical shift predictions, would have considerable discrepancies with those observed for altechromone A (see Figure 25).

### 2.12. Arunicin B

Woo and co-workers [50,51] isolated the monoterpenoid (+)-aruncin B (**37**) from the plant *Aruncusdioicus* var. *kamtschaticus*. The structure was assigned using various spectroscopic methods (NMR, MS, UV, and IR), and it was presumed to have an enol ether, a labile tertiary allylic ethyl ether, and a carboxylic acid functionality. However, the enol ether geometry was not established in the original paper. Ribaucourt and Hodgson [52] attempted a multi-step total synthesis of this enol ether, but all efforts to obtain the free acid from the Z-isomer (**Z-38**) led to rapid decomposition. The more stable E-isomer (**E-38**) was synthesized, but attempts to synthesize the E-Na salt also resulted in a similar decomposition. The authors hypothesized that Z-γ-alkylidenebutenolide (**39**) may be more consistent with the original reported data of **37**. To validate this hypothesis, they synthesized **39** and found that the NMR data matched those reported for the original aruncin B.



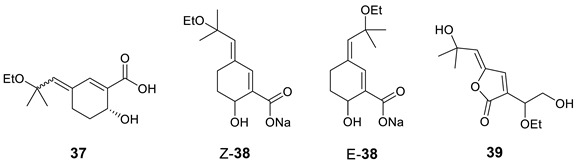



Results of the verification of originally proposed structure **37** using published ^13^C NMR chemical shifts [50] and predicted by ACD/SE program are shown in Figure 26 (experimental NMR spectra are presented in Appendix A).

As can be seen in Figure 26, the average and maximum deviations are large, suggesting that structure **37** cannot be correct. This conclusion is also supported by the IR spectrum of **37**, which shows a carbonyl absorption band at 1756 cm^−1^, contradicting the proposed structure **37** as that band is expected only in five-membered ring lactone molecules. To uncover the real structure of aruncin B using ACD/SE, we used the program option “create project for structure” because 2D NMR data were not available. The created MCD based on HSQC (Appendix A) was used for structure generation, which was completed with the following results: k = 20,918 → 10,740 → 4556, *t_g_* = 12 s. The three top-ranked structures are shown in Figure 27.

From Figure 27 we see that despite the absence of 2D NMR data, the program managed to establish the revised structure **39** as the correct one. The original structure occupies the 37th position in the output file.

## 3. Discussion

Structure elucidation of isolated natural products is typically accomplished using a variety of spectroscopic techniques, including NMR, MS, UV-IR, ECD, VCD, and others. Among these methods, NMR plays a key role because of its non-destructive nature, relatively high sensitivity, and its ability to provide detailed structural information. Despite the advancement of modern 1D and 2D NMR methods, errors can still exist when establishing the structure of complex molecules. This is not surprising given the nature of the spectral information from which the structure is derived. It has been shown [53] that this information has several shortcomings, including ambiguity of the standard length correlations (2 to 3 bonds) in HMBC and COSY spectra, which are difficult to differentiate one from the other without additional experiments; incomplete spectral information due to missing correlations; the existence of longer correlations (more than 3 bonds) that are difficult to confirm without additional DFT analysis; signal overlap in 1D and 2D spectra; and subjectivity in the specialist’s interpretation. With such limitations, manual interpretation of spectra may still result in erroneous structures being published in the future. However, the use of the CASE approach utilizing the ACD/SE program, as demonstrated in [54], can significantly reduce the number of erroneous structures in the literature and simplify their revision.

In this paper, we investigated twelve real examples of structure revision in which the misinterpretation of the original structures and the correctness of the proposed alternatives were proved by total synthesis (see summary of proposed and revised structures in Appendix A).

Our study has demonstrated that comparing experimental ^13^C NMR chemical shifts with those predicted by empirical methods implemented in the ACD/SE program enabled us to quickly verify proposed structures. This approach allowed us to identify all proposed structures as either questionable or undoubtedly incorrect, which took only a fraction of a second or a few seconds, replacing the need for the lengthy process of total synthesis. Once the original structure is proven incorrect, it becomes necessary to propose the correct structure or several possible candidates. However, given that experts may reach different conclusions, a systematic method of logically inferring all possible structures is essential. This is where expert computer systems come into play.

We have demonstrated that when an erroneous structure was established using 2D NMR spectra (HMBC and COSY), entering the spectral information into the program enables us to confidently determine the correct structure without the need for revision. This outcome was observed in all six cases where 2D NMR data were available. Notably, in three out of the six cases, the correct structure was identified using fuzzy structure generation, a method that allows for the solving of problems with an unknown number of non-standard correlations of unknown length. We have previously demonstrated [32] that verifying the correctness of the structure obtained through FSG can be accomplished by DFT calculations of the ^n^*J*_CH_ (*n* > 3) coupling constants, which are responsible for non-standard correlations identified during the generation process. These calculations validated structures of samoquasine A (which had one non-standard correlation) and dichomitol (which had two non-standard correlations).

As noted previously, for the other half of studied cases, the 2D NMR spectra were not available, making it more challenging to identify the correct structure. However, the ACD/SE expert system offers tools for structure analysis based on 1D NMR spectra alone. For instance, the “cutting out a green fragment” option can be applied effectively when a large fragment in which a good match between the experimental and calculated chemical shifts is obtained. The program creates an MCD in which this fragment and remaining free atoms are displayed (see, for instance, Appendix A). As a result, all generated structures contain this fragment. For example, in the case of palmuracin B6, the correct structure was reliably identified in 30 s using 1D spectra only. For norcarbenzoxazole G, two of the three empirical methods of chemical shift prediction indicated the correct structure, and then the revised structure was unequivocally confirmed by DFT calculations of chemical shifts.

Determining the correct structure can be even more challenging when the 2D NMR data are not available and the original structure does not contain a large green fragment. In such cases, the ACD/SE option “create project for structure” can be utilized. This option creates an artificial MCD based on an existing incorrect structure, and then the ACD/SE program carries out a search for all possible structures consistent with that MCD. There are two options when this method can be applied. 

First, the MCD is created from HSQC-type correlations meaning that the only constraints in the generation process are the hybridization of carbon atoms and the possibility of the direct connection of carbon atoms to heteroatoms (see Appendix A). This option can be applied only for moderate size molecules such as those with 14–17 skeletal atoms in our examples (see Section 2.10, Section 2.11 and Section 2.12). The structure generation time using this technique ranged from 12 s to 6 min for the problems encountered in Section 2.10, Section 2.11 and Section 2.12.

Second, the MCD is created from HSQC and all theoretically possible HMBC correlations corresponding to the originally proposed structure. This approach can be successful only providing that the standard correlation MCD of the erroneous structure is identical to that of the correct one. As seen from the analysis of hetiamicin A (Section 2.9), using this option, 60 structures were generated within 3 min 15 s and the correct structure of hetiamacin A was identified through standard chemical shift ranking and then validated by DFT calculation. The “create project for structure” method is lacking the possibility of incorrect correlation length determination, which is often used to filter out the erroneous structures, and, therefore, it is not recommended as a routine method for structure revisions. Better understanding of the limitations of this method will be the subject of our future studies.

The DFT calculations were used here for three purposes. First, to verify the correct structure based on chemical shift calculations when empirical methods of chemical shift prediction were incapable of differentiating structures in the ACD/SE output file (palmarumycin B6, norcarbenzoxazole G, hetiamacins A). Second, to verify non-standard correlations (^n^*J*_CH_, *n* > 3) using DFT computations of *J*-couplings when the FSG was utilized in the process of structure elucidation (dichomitol, samoquasine A). Third, to determine relative stereochemistry in the structures with chiral centers (dichomitol). The addition of DFT computations to the CASE expert system has a truly synergistic effect. It makes the CASE analysis of constitutional isomers more robust and expands the capabilities of CASE in solving stereochemical problems.

## 4. Materials and Methods

### 4.1. ACD/Structure Elucidator

In the present work, we used the ACD/Structure Elucidator (ACD/SE) expert system [55] as a primary tool for the verification and revision of misassigned structures. It has been described in many publications (for instance [53,56]), so here we will briefly explain only its main features. 

The ACD/SE program provides the option to utilize various constraints that can be applied to hypothetical structures based on molecular spectra and other sources of structural information (such as sample origin or chemical knowledge). A primary role in structure elucidation is played by NMR experimental data, which usually include 1D ^13^C and ^1^H, and 2D ^1^H-^13^C HMBC, ^1^H-^1^H COSY, ^1^H-^13^C HSQC NMR spectra. It is not necessary, but any other 2D NMR spectral data can be included too. The accurate molecular mass of the unknown compound, which is typically determined by high-resolution mass spectrometry (HRMS), should be entered into the program to establish its molecular formula. Information on some functional groups that is available from the IR/Raman and UV spectra can also be utilized.

The program processes raw 1D and 2D NMR spectra, performing peak picking and transferring chemical shifts, multiplicities, and coupling constants (if available) to a data table format. It then transforms ^1^H-^1^H COSY, ^1^H-^13^C HMBC (^1^H-^15^N HMBC, if available) correlations into connectivities between corresponding skeletal atoms. Using the molecular formula and connectivities, it creates a Molecular Connectivity Diagram (MCD) that captures the structural blocks of C, CH, CH_2_, CH_3_, NH, NH_2_, OH, and heteroatoms, as well as free hydrogens. The program annotates the structural blocks with experimental ^13^C and ^1^H chemical shifts and atom properties, such as hybridization and the possibility of neighboring with heteroatoms, using empirical rules captured by the program. The connectivities are visualized by arrows of different colors, with green representing HMBC correlations and blue representing COSY. The MCD window displays the information visually and can be edited by the operator. This information is used by the program to generate structures. 

The program offers a variety of options for structure generation, which can be selected by the user. These include strict generation, where all HMBC and COSY correlations are assumed to be of standard length (2–3 chemical bonds), and fuzzy structure generation (FSG), which allows for the presence of non-standard correlations (NSC) of unknown lengths. Generated structures are then filtered using both spectral (based on ^13^C and ^1^H characteristic chemical shifts) and structural criteria (using a library of unusual organic fragments). In the process of structure generation, the program predicts ^13^C chemical shifts for each structure using incremental and neural network algorithms with the speed of 10–30 thousand shifts per second on a regular laptop computer. The average deviations between the measured and predicted spectra are calculated and outputted in the file, which contains a set of structures. Duplicate structures are removed based on their average deviation values, with only the structure that had the minimum deviation being retained in the file as the best representative of the family. Depending on the number and rigidity of the constraints imposed by the spectroscopic data and additional information, the output file may contain anywhere from one to thousands of structures. We used the following notations for the structure generation and filtration process: k = 1000 → 100 → 10, *t_g_* = 5 s,
where k is the number of generated structures (1000), 100 is the number of retained structures after filtration, 10 is the number of retained structures after filtration and duplicate removal, and *t_g_* (5 s) is the CPU time of structure generation. 

To determine the most probable structure, the output file is sorted in ascending order of the average deviation value, d_N_. This procedure typically results in the correct structure being ranked first or near the top of the file. Next, ^13^C NMR spectra are computed for the top 10 to 50 structures in the sorted file using a fragmental method that employs the Hierarchical Organization of Spherical Environments (HOSE) code approach [57]. The corresponding average deviation values (d_A_) are then computed, and the structures are re-sorted according to their d_A_ values. The HOSE code-based approach is marginally more accurate than neural networks. By performing additional rounds of ranking, the probability of identifying the correct structure as the top-ranked structure is increased. Notably, it has been observed that in the vast majority of cases, the correct structure exhibits the minimum values of average deviation as determined by all three methods.

When ACD/SE is utilized for structure verification, the following two auxiliary modes are particularly helpful: (a) “create project for structure” generates a molecular connectivity diagram (MCD) that includes all theoretical correlations corresponding to the kinds of 2D NMR spectra (HSQC, HMBC, etc.) which were specified by the user; (b) “cutting out a green fragment” produces an MCD that contains a fragment selected by the user in a structure with assigned chemical shifts. Both these procedures were utilized to revise structures discussed in our article.

The methodology outlined here enables the frequent selection of the most probable structure, as well as the identification of a minimal set of plausible structures for further analysis using DFT-based chemical shift and *J*-coupling predictions [15,32,58].

### 4.2. DFT Calculations

Chemical shift analysis using DFT calculations were usually done for three to six top-ranked candidates from the ACD/SE output file. For the molecule with rotatable bonds, prior to chemical shift calculations, the conformational analysis of all top-ranked candidates was carried out. For this purpose, we usually used either the MMFF94 force-field computations in Spartan’20 [59] or the OPLS3e force-field computations in MacroModel [60]. Then, conformational ensembles were DFT-optimized using the Gaussian16 program [61]. Chemical shift calculations were done either with Spartan’20 or the Gaussian16 program, as previously described [15,32,58]. Conformationally averaged ^13^C and ^1^H chemical shifts were obtained by using a Boltzmann distribution based on DFT electronic energies. Averaged chemical shift values were then compared with the experimental values by computing root-mean-square deviations (RMSD), maximum deviations (max_dev), and correlation coefficients (*r*) with the Excel program [62]. More details about the choice of DFT functionals and basis sets can be found in the Appendix A. *J*-coupling analysis was done using DFT computations at the B3LYP/6-311+G(d,p)-mixed//B3LYP/6-31G(d) level of theory on the Gaussian16 program, as previously described [32].

## 5. Conclusions

In conclusion, we have proposed a methodology that allows the revision of structures without the need for synthesis. Starting with verification of the erroneously proposed initial structure, our proposed workflow will proceed with full structure elucidation and the establishment of the correct structure using the original data. A combination of the CASE and DFT methods provides the opportunity not only to determine the correct constitutional isomer, but also the relative configuration at the chiral centers, if they are part of the molecule. While total synthesis will still have an important role and will continue to be used as a reliable means for structural elucidation of natural products, it is often tedious, sometimes challenging, time-consuming, and resource intensive, as shown in several case studies illustrated in this paper. Our approach not only reduces the labor and material required to perform the structure revision, but also demonstrates how to prevent the misinterpretation and publication of incorrect structures, while providing significant time saving which can be invaluable to researchers.

## Figures and Tables

**Figure 1 molecules-28-03796-f001:**
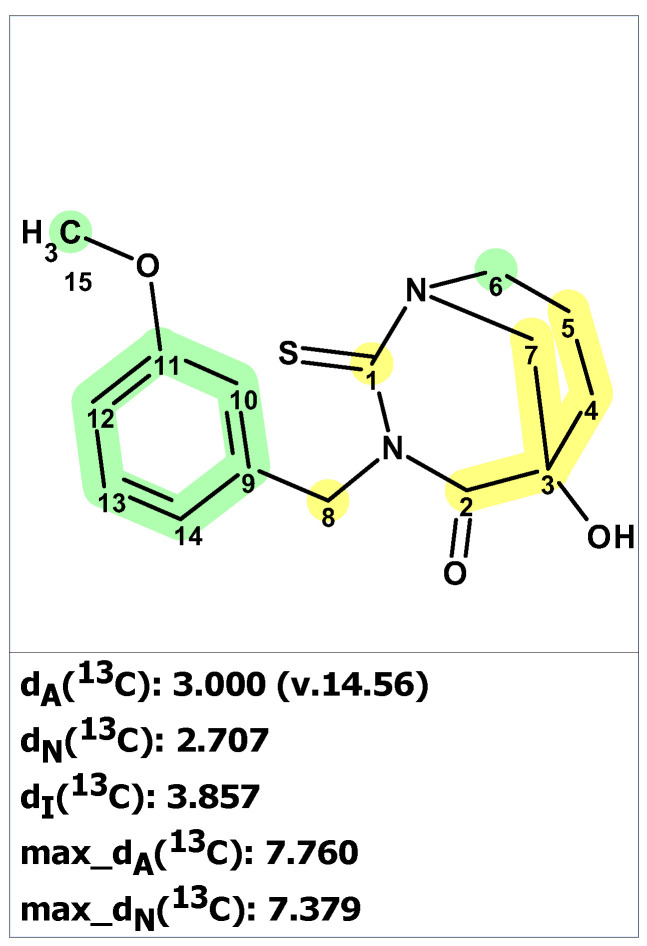
Proposed structure of macahydantoin B [18] for which ^13^C chemical shift predictions were carried out. The average deviations of the ^13^C chemical shifts determined by methods A, N, and I are denoted as dA, dN, and dI, respectively. Each atom is colored to indicate the difference between its experimental and calculated ^13^C chemical shifts. Green color represents a difference between 0 to 3 ppm, yellow indicates a difference greater than 3 ppm but less than 15 ppm.

**Figure 2 molecules-28-03796-f002:**
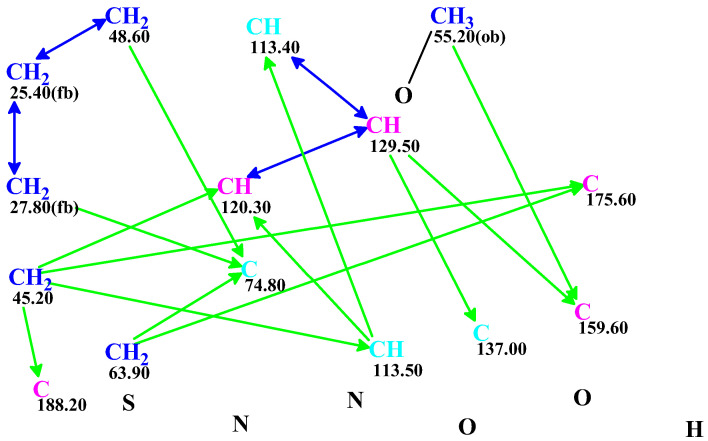
Molecular connectivity diagram (MCD) for macahydantoin B based on COSY (blue arrows) and HMBC (green arrows) correlations. The hybridizations of carbon atoms are marked by corresponding colors: *sp2*—violet, *sp3*—blue, *sp2* or *sp3*—light blue. Labels “ob” and “fb” are set by the program to carbon atoms, for which neighboring with heteroatom is either obligatory (ob) or forbidden (fb).

**Figure 3 molecules-28-03796-f003:**
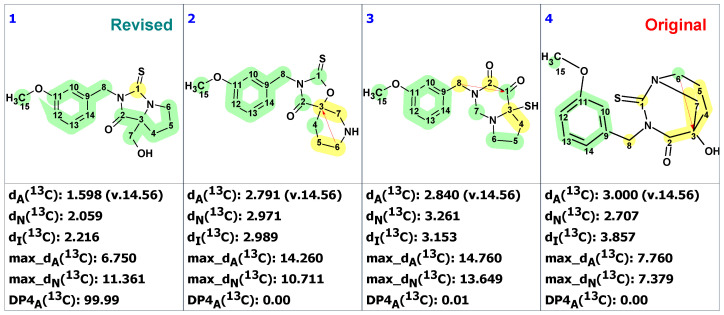
Four top-ranked structures from the output file of the CASE analysis of macahydantoin B. Red arrows indicate nonstandard HMBC correlations (NSC)—those whose lengths exceed three chemical bonds (^n^*J*_CH_*, n* > 3). Each atom is colored to indicate the difference between its experimental and calculated ^13^C chemical shifts. Green color represents a difference between 0 to 3 ppm, yellow indicates a difference greater than 3 ppm but less than 15 ppm.

**Figure 4 molecules-28-03796-f004:**
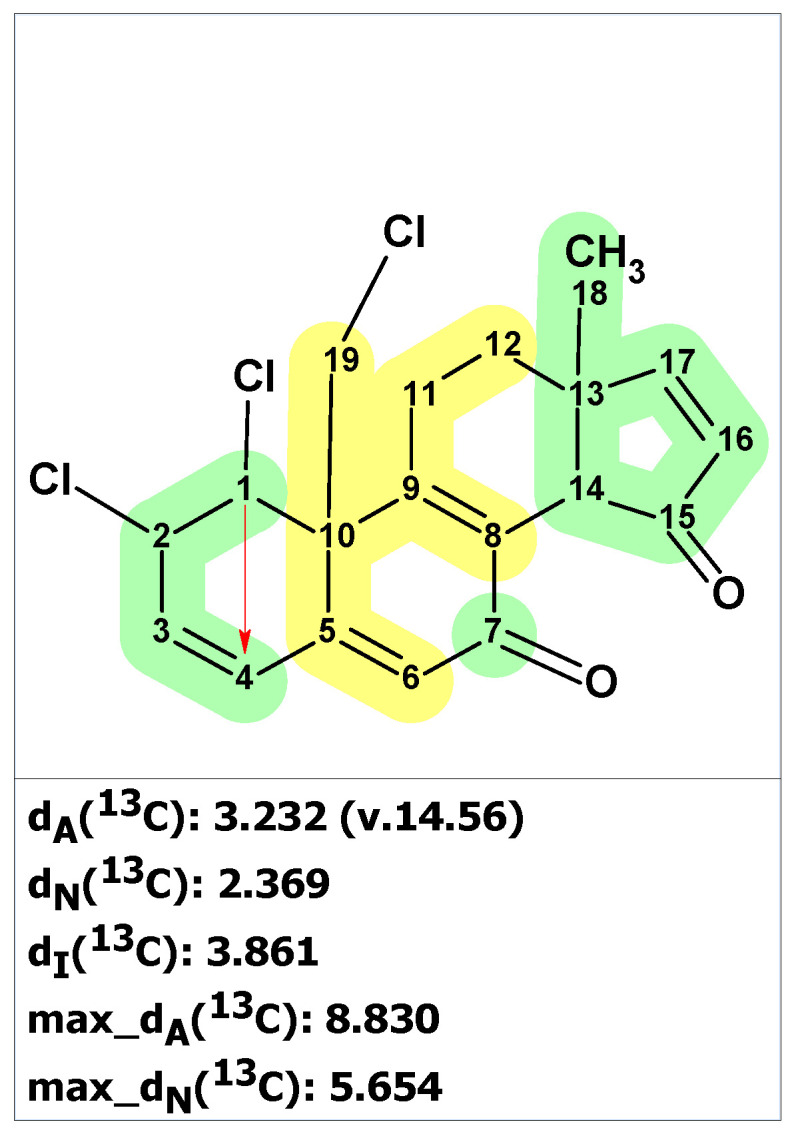
Structure of clionastatin A determined by ACD/SE. Red arrows indicate nonstandard HMBC correlations (NSC)—those whose lengths exceed three chemical bonds (^n^*J*_CH_*, n* > 3). Each atom is colored to indicate the difference between its experimental and calculated ^13^C chemical shifts. Green color represents a difference between 0 to 3 ppm, yellow indicates a difference greater than 3 ppm but less than 15 ppm.

**Figure 5 molecules-28-03796-f005:**
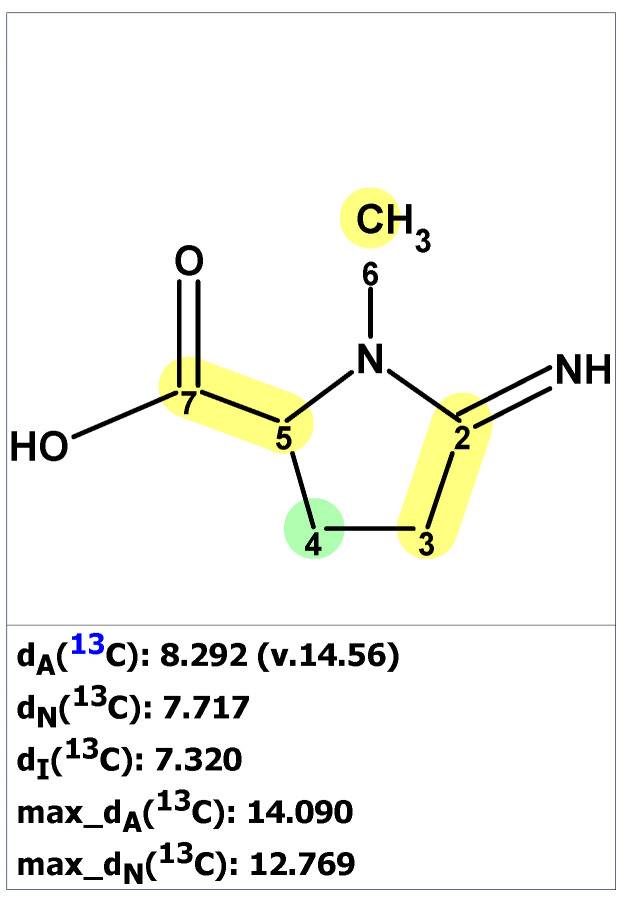
Results of ^13^C chemical shift prediction for originally proposed structure pyrostatin B (**6**). Each atom is colored to indicate the difference between its experimental and calculated ^13^C chemical shifts. Green color represents a difference between 0 to 3 ppm, yellow indicates a difference greater than 3 ppm but less than 15 ppm.

**Figure 6 molecules-28-03796-f006:**
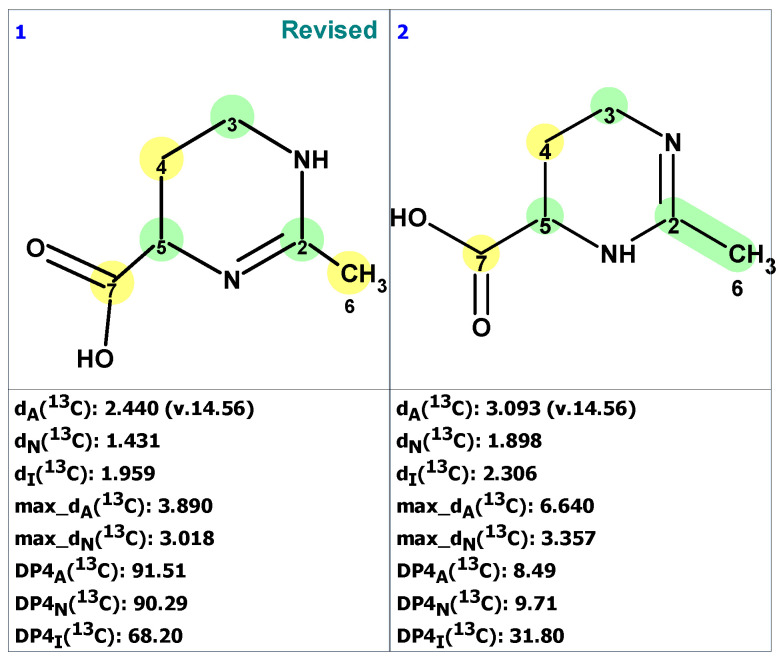
Two top-ranked structures from the output file of the CASE analysis of pyrostatins B. Each atom is colored to indicate the difference between its experimental and calculated ^13^C chemical shifts. Green color represents a difference between 0 to 3 ppm, yellow indicates a difference greater than 3 ppm but less than 15 ppm.

**Figure 7 molecules-28-03796-f007:**
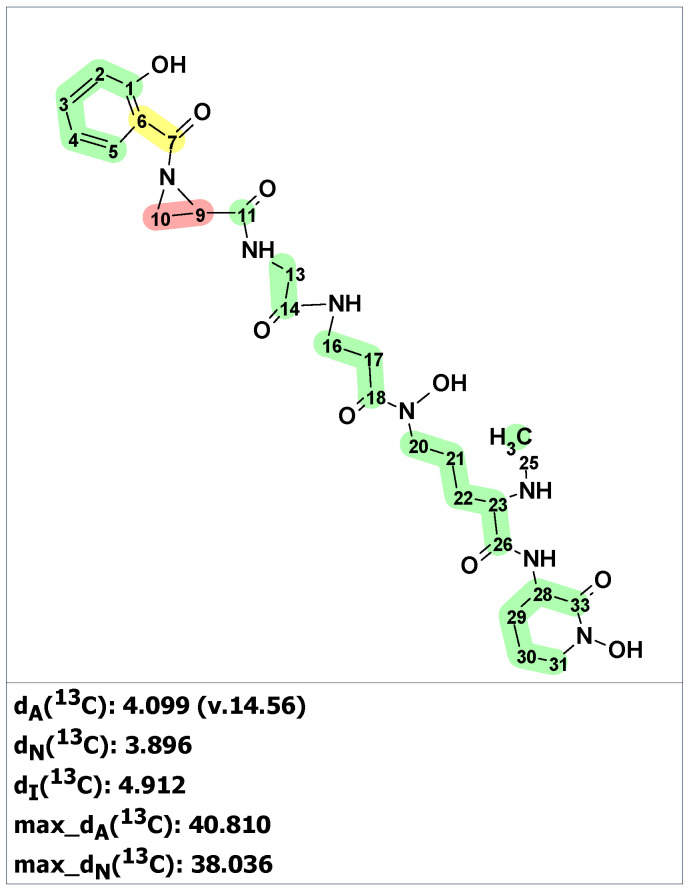
Results of ^13^C chemical shift prediction for originally proposed structure of madurastatin C (**9**) by ACD/SE. Each atom is colored to indicate the difference between its experimental and calculated ^13^C chemical shifts. Green color represents a difference between 0 to 3 ppm, yellow indicates a difference greater than 3 ppm but less than 15 ppm. The red color represents a difference between experimental and calculated chemical shifts which exceeds 15 ppm.

**Figure 8 molecules-28-03796-f008:**
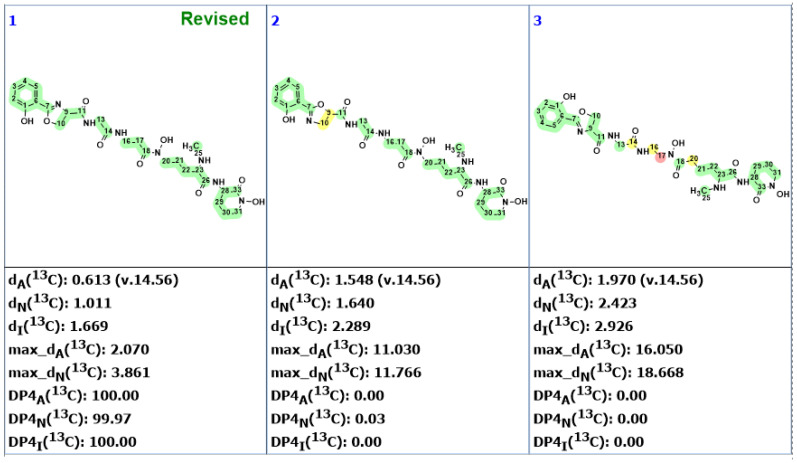
Three top-ranked structures from the output file of the CASE analysis of madurastatin C. Each atom is colored to indicate the difference between its experimental and calculated ^13^C chemical shifts. Green color represents a difference between 0 to 3 ppm, yellow indicates a difference greater than 3 ppm but less than 15 ppm. The red color represents a difference between experimental and calculated chemical shifts which exceeds 15 ppm.

**Figure 9 molecules-28-03796-f009:**
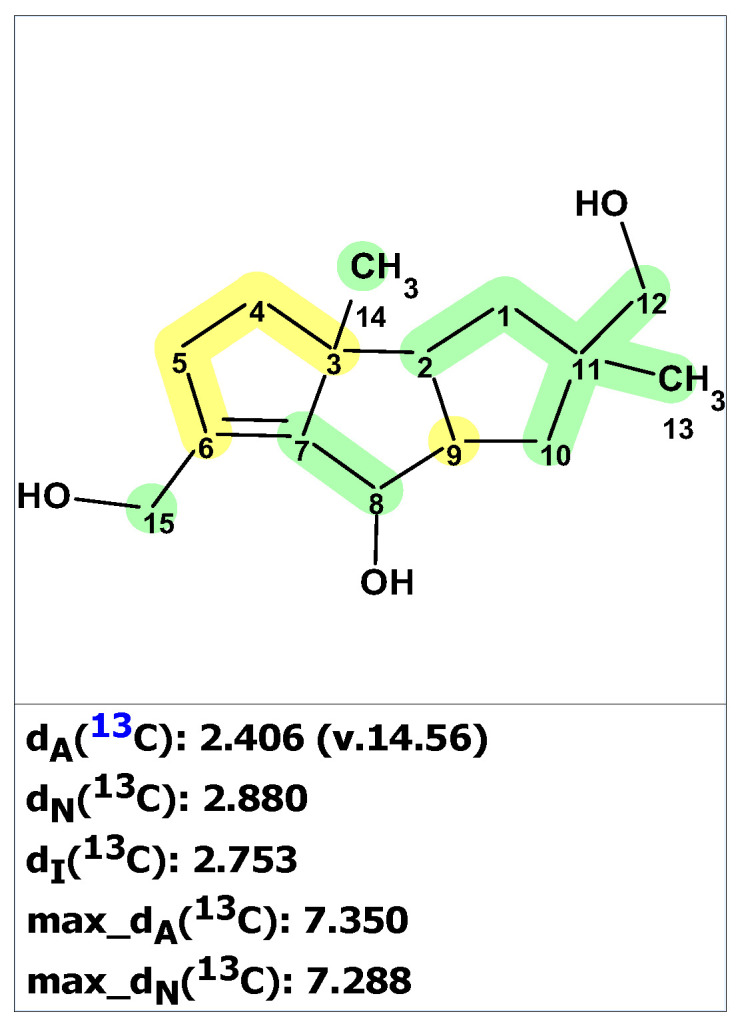
The results of ^13^C chemical shift prediction for originally proposed structure of dichomitol (**13**). Each atom is colored to indicate the difference between its experimental and calculated ^13^C chemical shifts. Green color represents a difference between 0 to 3 ppm, yellow indicates a difference greater than 3 ppm but less than 15 ppm.

**Figure 10 molecules-28-03796-f010:**
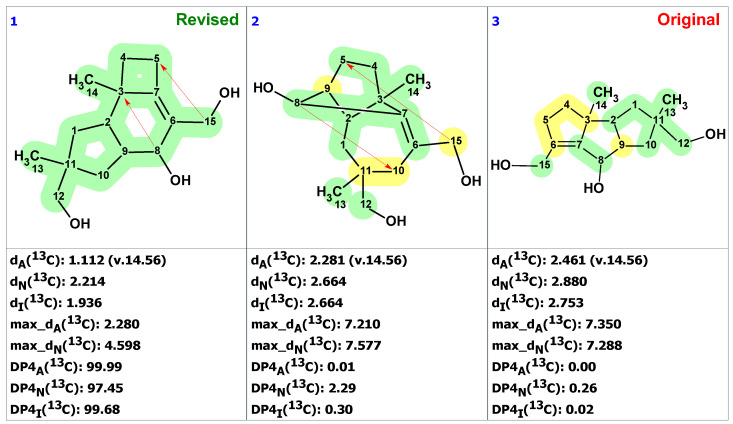
Three top-ranked structures from the output file of the CASE analysis of dichomitol. Red arrows indicate nonstandard HMBC correlations (NSC)—those whose lengths exceed three chemical bonds (^n^*J*_CH_*, n* > 3). Each atom is colored to indicate the difference between its experimental and calculated ^13^C chemical shifts. Green color represents a difference between 0 to 3 ppm, yellow indicates a difference greater than 3 ppm but less than 15 ppm.

**Figure 11 molecules-28-03796-f011:**
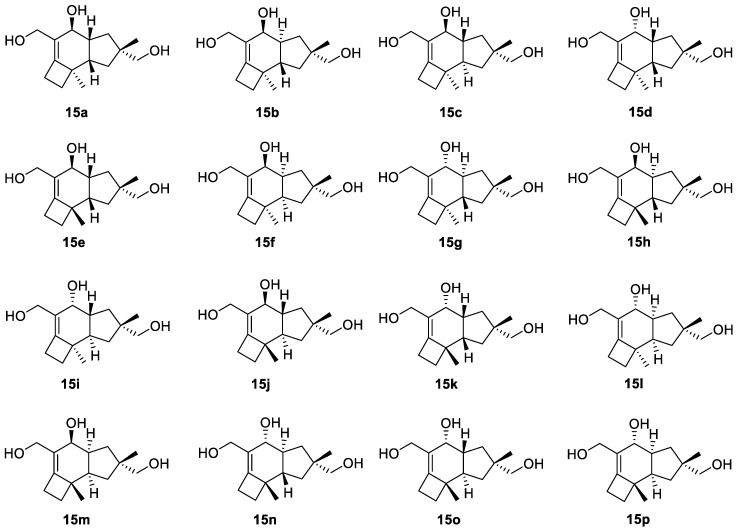
Sixteen possible diastereomers of dichomitol with fixed R-chirality at the C11 carbon. Natural diastereomer of dichomitol is **15a**.

**Figure 12 molecules-28-03796-f012:**
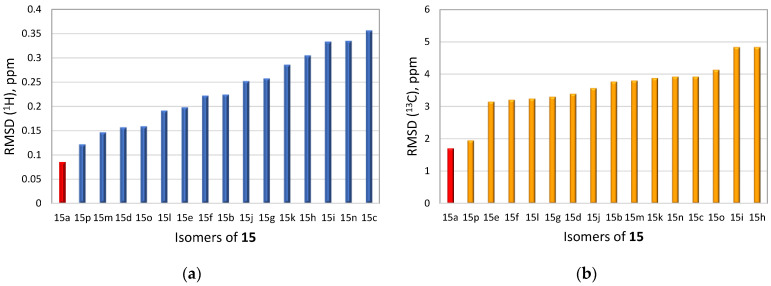
RMSD values for the DFT-calculated ^1^H (**a**) and ^13^C (**b**) chemical shifts of sixteen diastereomers of dichomitol (**15a**–**15p**). The RMSD values for the natural diastereomer of dichomitol **15a** are highlighted in red. DFT calculations were done at the mPW1PW91/6-311+G(2d,p)//B3LYP/6-31+G(d,p) level of theory (for more details see Appendix A).

**Figure 13 molecules-28-03796-f013:**
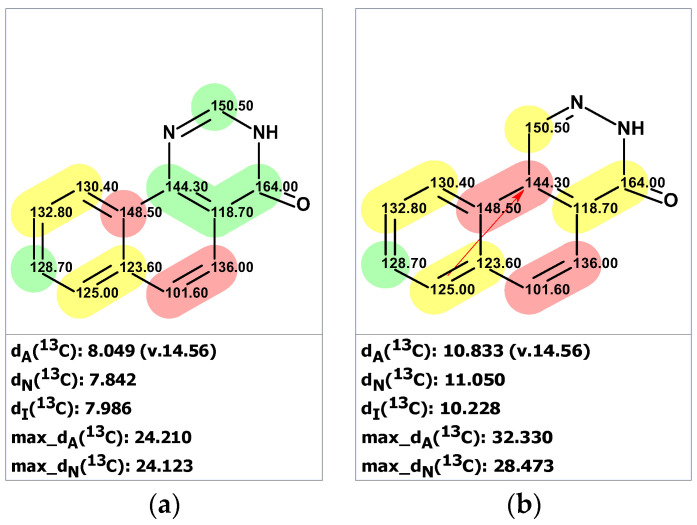
Results of ^13^C chemical shift prediction for structure **16** (**a**) and for structure **18** (**b**). Red arrows indicate nonstandard HMBC correlations (NSC)—those whose lengths exceed three chemical bonds (^n^*J*_CH_*, n* > 3). Each atom is colored to indicate the difference between its experimental and calculated ^13^C chemical shifts. Green color represents a difference between 0 to 3 ppm, yellow indicates a difference greater than 3 ppm but less than 15 ppm. The red color represents a difference between experimental and calculated chemical shifts which exceeds 15 ppm.

**Figure 14 molecules-28-03796-f014:**
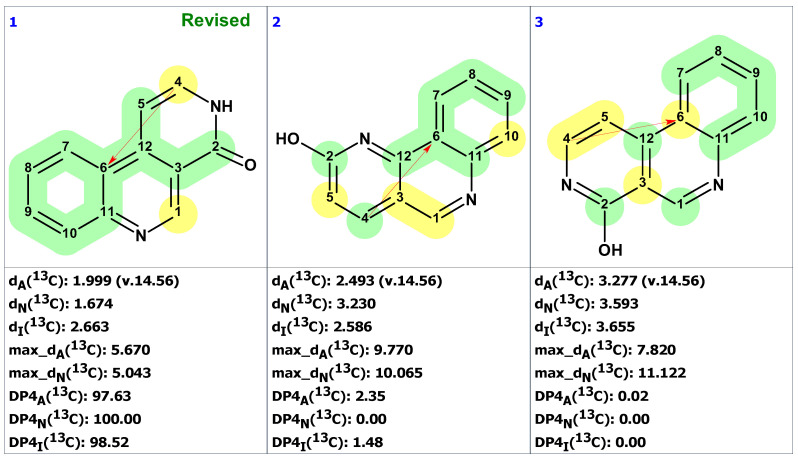
Three top-ranked structures from the output file of the CASE analysis of samoquasine A. Red arrows indicate nonstandard HMBC correlations (NSC)—those whose lengths exceed three chemical bonds (^n^*J*_CH_*, n* > 3). Each atom is colored to indicate the difference between its experimental and calculated ^13^C chemical shifts. Green color represents a difference between 0 to 3 ppm, yellow indicates a difference greater than 3 ppm but less than 15 ppm.

**Figure 15 molecules-28-03796-f015:**
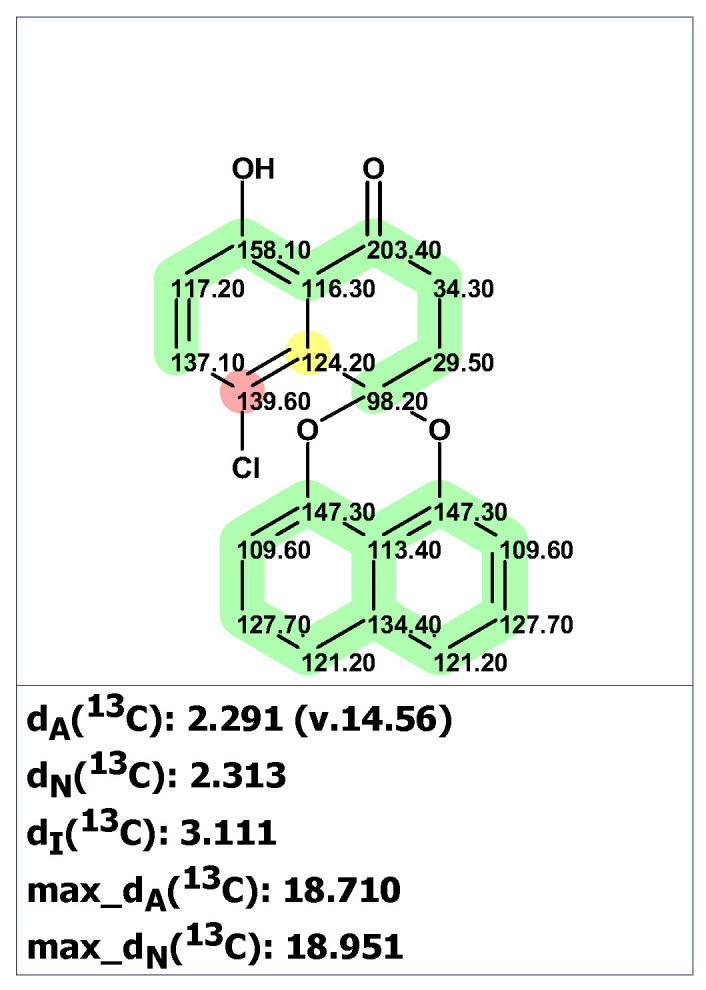
Results of the ^13^C chemical shift prediction for the originally proposed structure of palmarumycin B6 (**19**). Red arrows indicate nonstandard HMBC correlations (NSC)—those whose lengths exceed three chemical bonds (^n^*J*_CH_*, n* > 3). Each atom is colored to indicate the difference between its experimental and calculated ^13^C chemical shifts. Green color represents a difference between 0 to 3 ppm, yellow indicates a difference greater than 3 ppm but less than 15 ppm. The red color represents a difference between experimental and calculated chemical shifts which exceeds 15 ppm.

**Figure 16 molecules-28-03796-f016:**
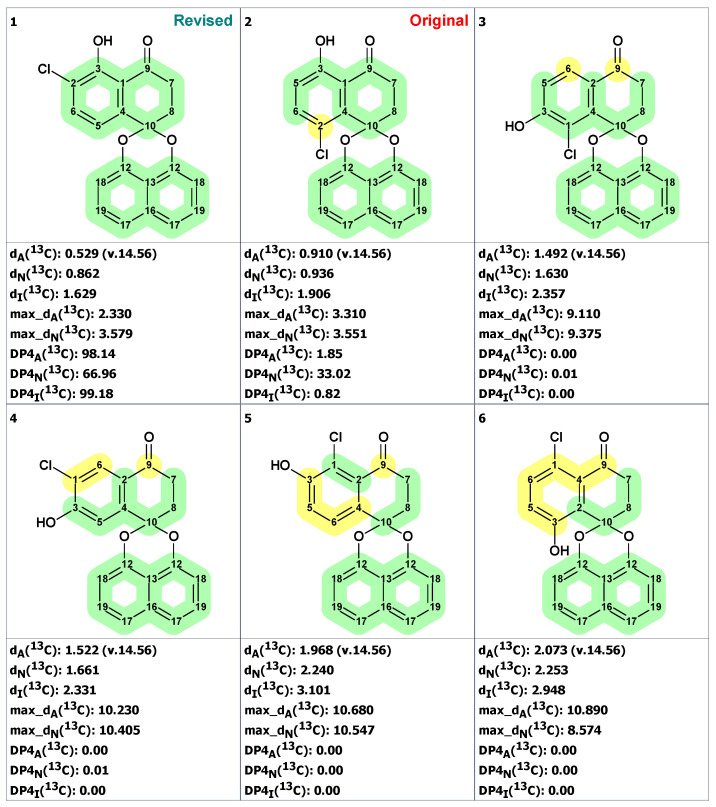
Six top-ranked structures from the output file of the CASE analysis of palmarumycin B6. Each atom is colored to indicate the difference between its experimental and calculated ^13^C chemical shifts. Green color represents a difference between 0 to 3 ppm, yellow indicates a difference greater than 3 ppm but less than 15 ppm.

**Figure 17 molecules-28-03796-f017:**
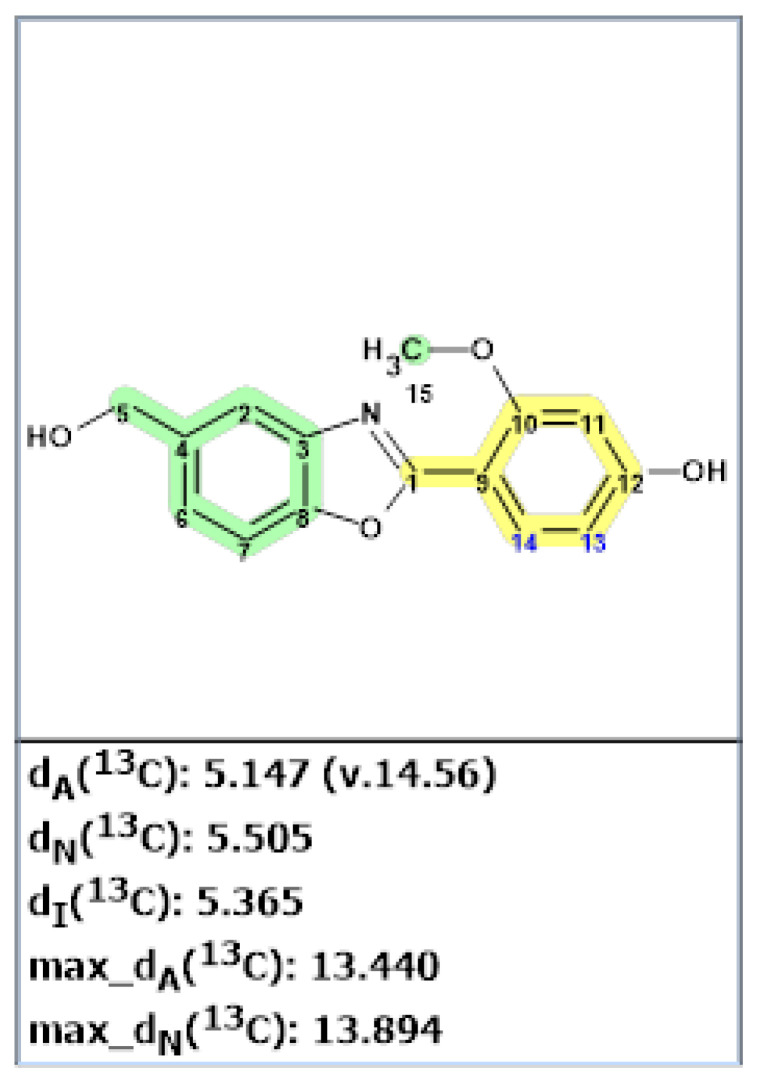
Results of ^13^C chemical shift prediction for the originally proposed structure of norcarbenzoxazole G (**22**). Each atom is colored to indicate the difference between its experimental and calculated ^13^C chemical shifts. Green color represents a difference between 0 to 3 ppm, yellow indicates a difference greater than 3 ppm but less than 15 ppm.

**Figure 18 molecules-28-03796-f018:**
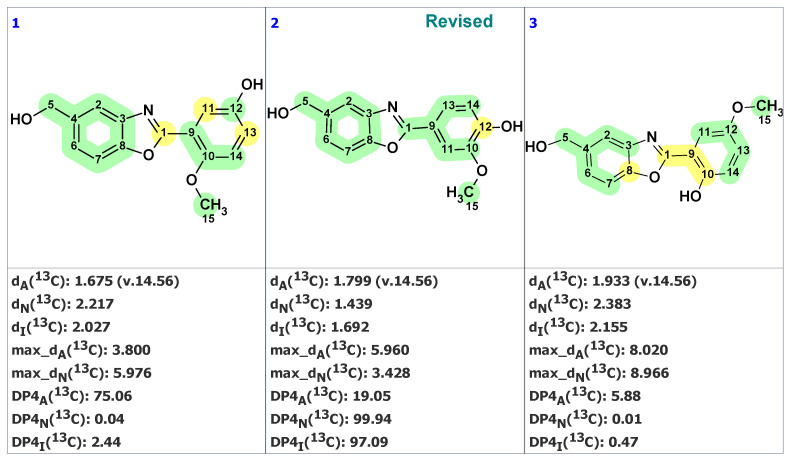
Three top-ranked structures from the output file of the CASE analysis of norcarbenzoxazole G. Each atom is colored to indicate the difference between its experimental and calculated ^13^C chemical shifts. Green color represents a difference between 0 to 3 ppm, yellow indicates a difference greater than 3 ppm but less than 15 ppm.

**Figure 19 molecules-28-03796-f019:**
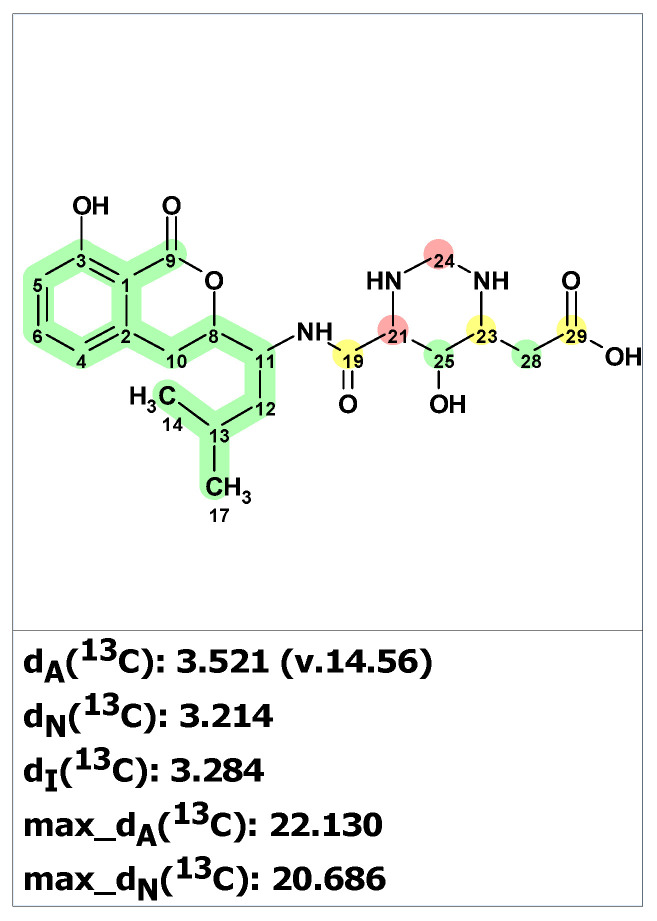
Results of ^13^C chemical shift prediction for the proposed structure hetiamacin A (**24**). Each atom is colored to indicate the difference between its experimental and calculated ^13^C chemical shifts. Green color represents a difference between 0 to 3 ppm, yellow indicates a difference greater than 3 ppm but less than 15 ppm. The red color represents a difference between experimental and calculated chemical shifts which exceeds 15 ppm.

**Figure 20 molecules-28-03796-f020:**
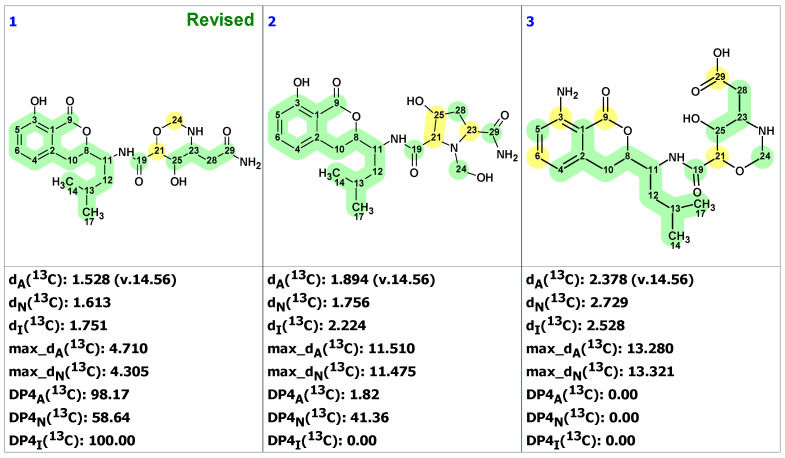
Three top-ranked structures from the output file of the CASE analysis of hetiamacin A. Each atom is colored to indicate the difference between its experimental and calculated ^13^C chemical shifts. Green color represents a difference between 0 to 3 ppm, yellow indicates a difference greater than 3 ppm but less than 15 ppm.

**Figure 21 molecules-28-03796-f021:**
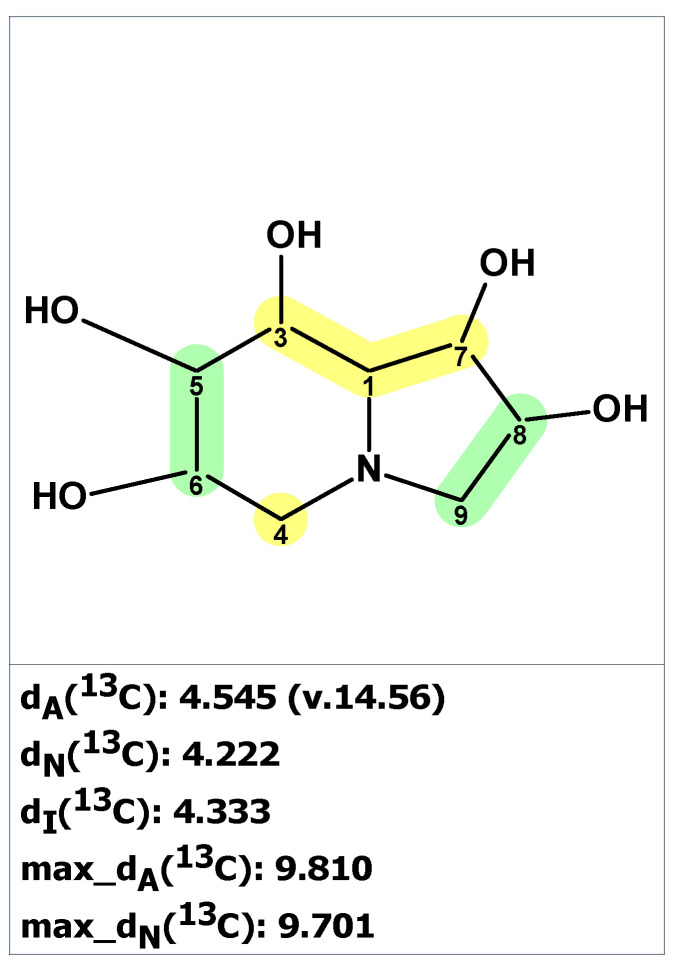
Statistical analysis of ^13^C chemical shift prediction for the original structure of uniflorine A (**27**) by ACD/SE. Each atom is colored to indicate the difference between its experimental and calculated ^13^C chemical shifts. Green color represents a difference between 0 to 3 ppm, yellow indicates a difference greater than 3 ppm but less than 15 ppm.

**Figure 22 molecules-28-03796-f022:**
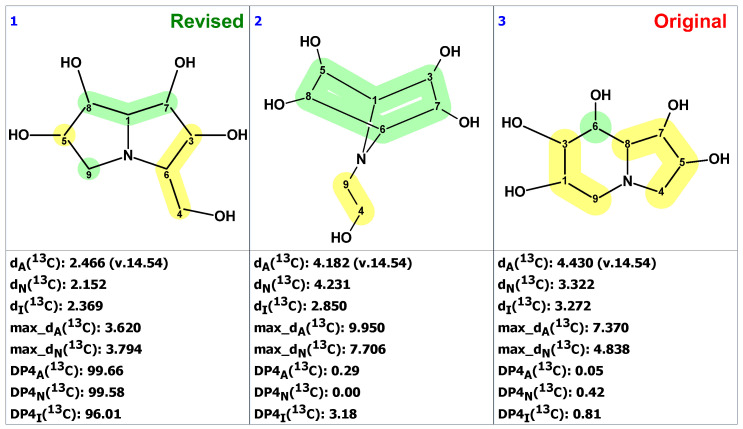
Three top-ranked from the output file of the CASE analysis of uniflorine A. Each atom is colored to indicate the difference between its experimental and calculated ^13^C chemical shifts. Green color represents a difference between 0 to 3 ppm, yellow indicates a difference greater than 3 ppm but less than 15 ppm.

**Figure 23 molecules-28-03796-f023:**
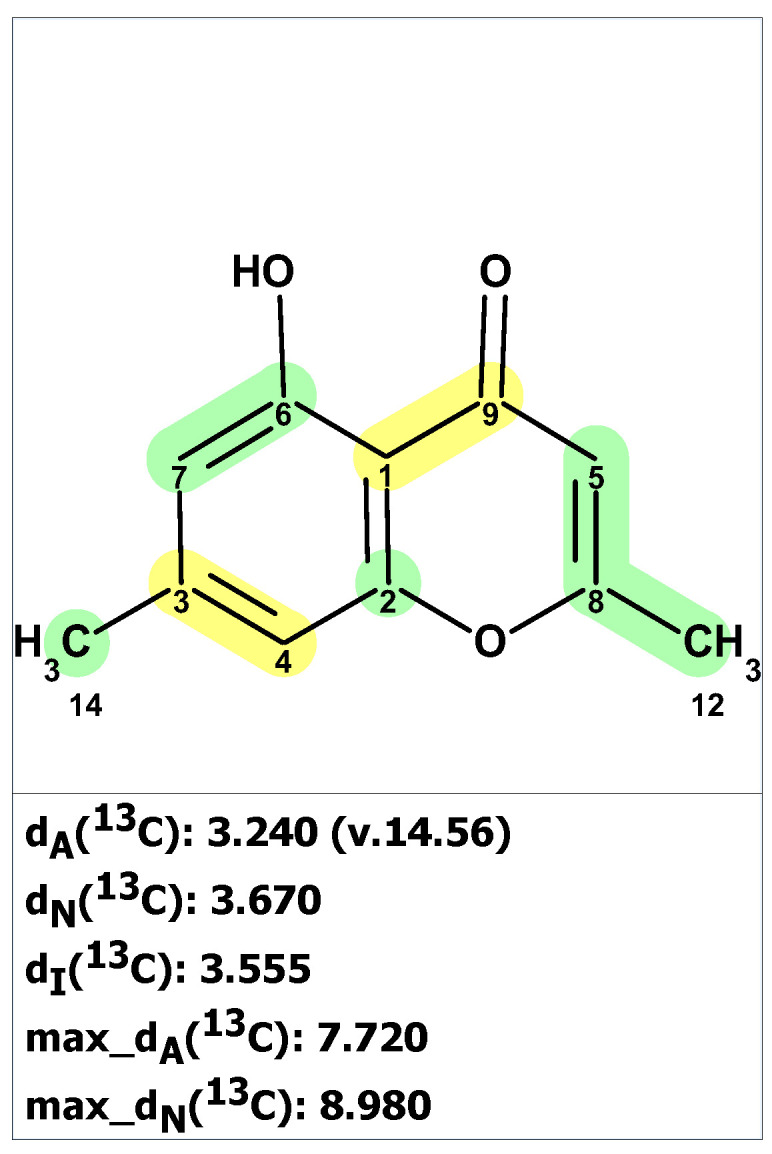
Results of ^13^C chemical shift prediction for the original structure of altechromone A (**34**). Each atom is colored to indicate the difference between its experimental and calculated ^13^C chemical shifts. Green color represents a difference between 0 to 3 ppm, yellow indicates a difference greater than 3 ppm but less than 15 ppm.

**Figure 24 molecules-28-03796-f024:**
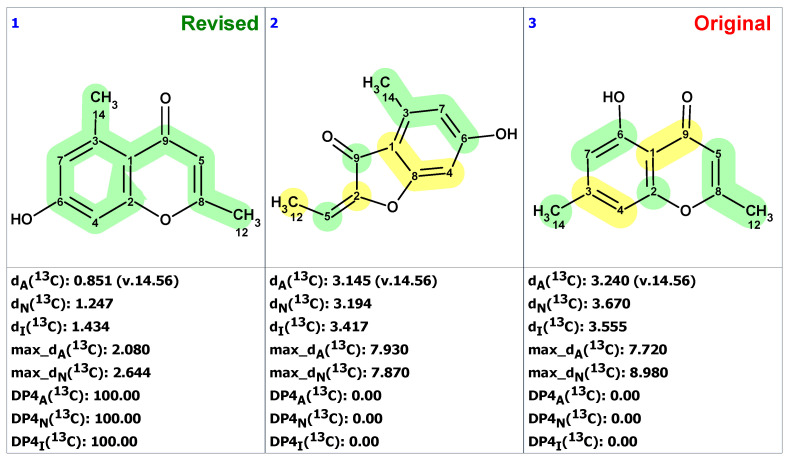
Three top-ranked structures from the output file of the CASE analysis of altechromone A. Each atom is colored to indicate the difference between its experimental and calculated ^13^C chemical shifts. Green color represents a difference between 0 to 3 ppm, yellow indicates a difference greater than 3 ppm but less than 15 ppm.

**Figure 25 molecules-28-03796-f025:**
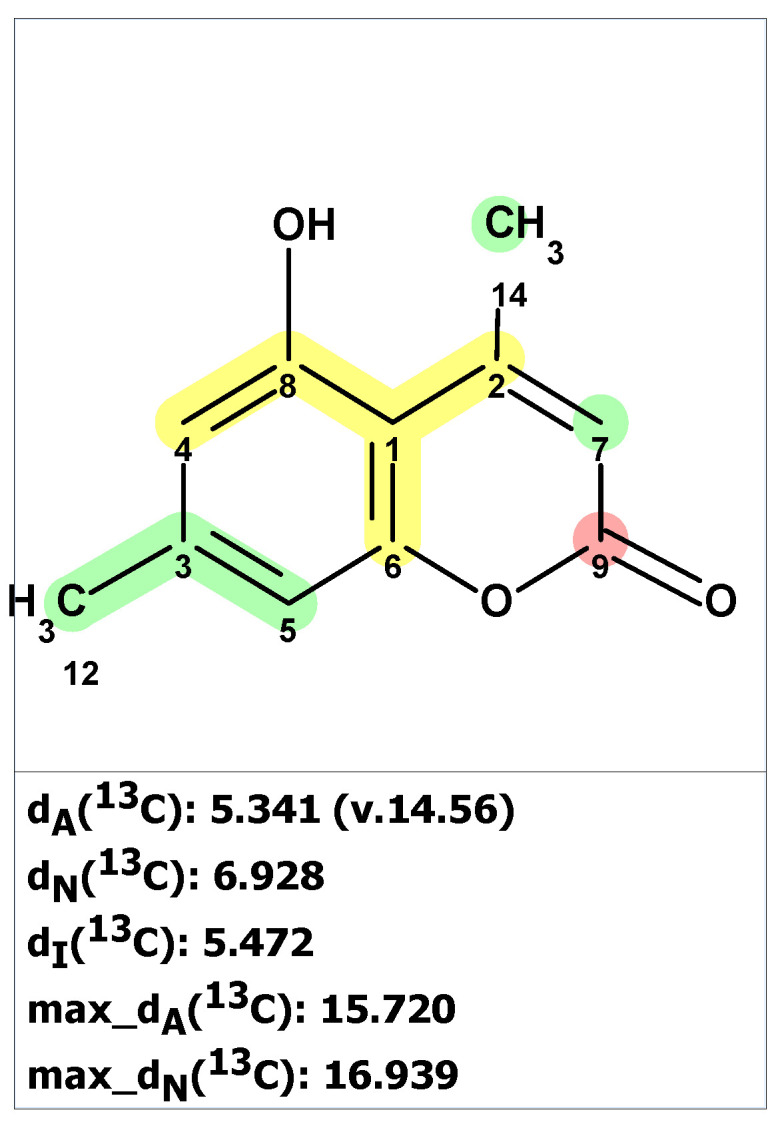
^13^C chemical shifts deviations between predicted by ACD/SE for structure **36** and experimental values for altechromone A [49]. Each atom is colored to indicate the difference between its experimental and calculated ^13^C chemical shifts. Green color represents a difference between 0 to 3 ppm, yellow indicates a difference greater than 3 ppm but less than 15 ppm. The red color represents a difference between experimental and calculated chemical shifts which exceeds 15 ppm.

**Figure 26 molecules-28-03796-f026:**
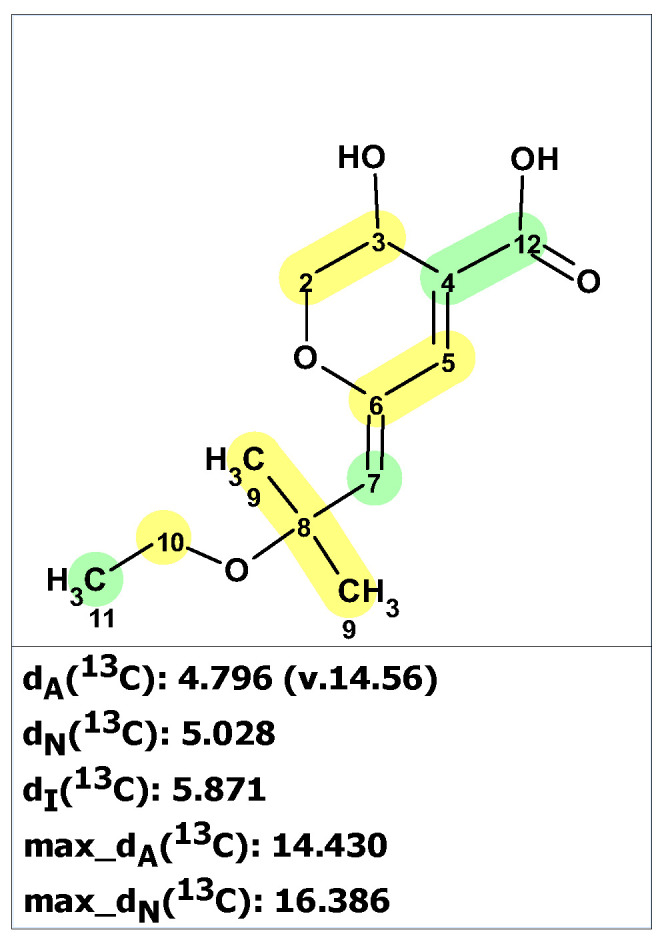
Results of ^13^C NMR chemical shift predictions for the original structure of aruncin B (**37)**. Each atom is colored to indicate the difference between its experimental and calculated ^13^C chemical shifts. Green color represents a difference between 0 to 3 ppm, yellow indicates a difference greater than 3 ppm but less than 15 ppm.

**Figure 27 molecules-28-03796-f027:**
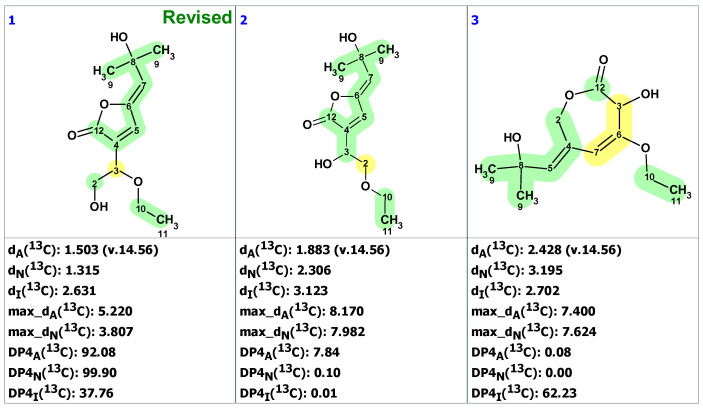
Three top-ranked structures from the output file of the CASE analysis of aruncin B. Each atom is colored to indicate the difference between its experimental and calculated ^13^C chemical shifts. Green color represents a difference between 0 to 3 ppm, yellow indicates a difference greater than 3 ppm but less than 15 ppm.

**Table 1 molecules-28-03796-t001:** Summary of DFT analysis of ^13^C NMR chemical shifts of the top six CASE-generated structures for palmarumycin B6 (see more details in Appendix A) *.

	Structure #1 (20)	Structure #2 (19)	Structure #3	Structure #4	Structure #5	Structure #6
RMSD, ppm	1.11	1.75	2.73	2.85	3.62	4.06
max_dev, ppm	2.8	4.3	8.7	9.1	9.3	9.8
*r*	0.9997	0.9991	0.9979	0.9974	0.9959	0.9944

* RMSD—root-mean-square deviation, max_dev—maximum deviation, *r*—correlation coefficient. DFT analysis was done at the ωB97-D/6-31G(d)//ωB97-D/6-31G(d) level of theory by Spartan’20 program.

**Table 2 molecules-28-03796-t002:** Summary of DFT analysis of ^13^C NMR chemical shifts of the top three CASE-generated structures and originally proposed structure **22** for norcarbenzoxazole G (for more details see Appendix A) *.

	Structure #1	Structure #2 (23)	Structure #3	Original (22)
RMSD, ppm	2.71	2.01	2.39	8.69
max_dev, ppm	4.5	3.4	6.5	17.4
*r*	0.9958	0.9981	0.9968	0.9600

* RMSD—root-mean-square deviation, max_dev—maximum deviation, *r*—correlation parameter. DFT analysis was done at the mPW1PW91/6-311+G(2d,p)//B3LYP/6-31+G(d,p) level of theory.

**Table 3 molecules-28-03796-t003:** Summary of DFT analysis of ^13^C NMR chemical shifts of the top three CASE generated structures and originally proposed structure **24** for hetiamacin A (for more details see Appendix A) *.

	Structure #1 (25)	Structure #2	Structure #3	Original (24)
RMSD, ppm	1.59	3.75	4.04	6.87
max_dev, ppm	3.3	9.6	10.9	22.2
*r*	0.9996	0.9976	0.9977	0.9927

* RMSD = root-mean-square deviation, max_dev—maximum deviation, *r*—correlation parameter. DFT analysis was done at the mPW1PW91/6-311+G(2d,p)//B3LYP/6-31+G(d,p) level of theory.

## Data Availability

The data presented in this study are available upon request from the corresponding authors.

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
