# Peer review of "Enhancing Efficiency of Natural Product Structure Revision: Leveraging CASE and DFT over Total Synthesis"

_molecules, 2023, doi:10.3390/molecules28093796_

Round 1
Reviewer 1 Report
In this paper, the authors present using of CASE protocol combined with DFT calculations of NMR chemical shifts as a useful tool for establishing the exact structure from experimentally available spectroscopic data.
The effectiveness of their methodology is presented through great number of interesting examples from the literature. Each time discussed methodology has proven to be a useful tool in assigning the correct structure.
The article is very nicely written and shows how computational methods can be an extremely useful tool in organic synthesis as well as in determining the structure of natural products without the need for total synthesis to confirm the structure
Author Response
We are grateful to reviewer #1 for taking time to review our manuscript and for positive feedback.
Reviewer 2 Report
1. The figure 7 is poor, pls improve it.
2. The solution to the structure revision problem can be found in the application of artificial intelligent (AI) technologies, which have been developing for the last three decades by a number of groups, this part should be cited the refs, such as J. Org. Chem. 2019, 84, 14627−14635; Org. Chem. Front., 2021, 8, 4554–4559; Monatsh Chem, 2017, 48,1259–1267 and J. Phys. Chem. A, 2019, 123, 6751−6760.
3. Pls highlight the introduction of this part.
4. I suggest the authors list a Table for such revision
it could be revised.
Reviewer 3 Report
Despite great care, specific structures can be established incorrectly. Ignoring some unusual chemical shifts or correlations in two-dimensional NMR spectra, which is very easy in the case of complex organic compounds, leads to confirmation of the wrong structure. In the paper, the authors applied Computer-Assisted Structure Elucidation (CASE) with a combination of Density Functional Theory calculation of NMR parameters (mainly 1H and 13C chemical shifts) as tools for verification of the initially proposed structure. The CASE algorithms are based on two major processes: a) high efficiency of structure generation and b) high accuracy and efficiency of prediction of NMR chemical shifts for generated structures. As an example, the authors corrected the structures of twelve known compounds and established their correct structure.
The proposed approach was satisfactorily tested and appeared helpful in elucidating and verifying complex organic compounds, such as natural products. The presentation of the material is clear and logical, and there are no significant errors in the presentation. I recommend publishing these paper as it is.
